# Modified Stückelberg Formalism: Free Massive Abelian 2-Form Theory in 4D

A. K. Rao [1] and R. P. Malik [1,2,*]

1 Physics Department, Institute of Science, Banaras Hindu University, Varanasi 221 005, India
2 DST Centre for Interdisciplinary Mathematical Sciences, Institute of Science, Banaras Hindu University, Varanasi 221 005, India
* Correspondence: rpmalik1995@gmail.com

**Abstract:** We demonstrate that the celebrated Stückelberg formalism is *modified* in the case of a *massive* four (3 + 1)-dimensional (4D) Abelian 2-form theory due to the presence of a self-duality *discrete* symmetry in the theory. The *latter* symmetry entails upon the *modified* 4D massive Abelian 2-form gauge theory to become a *massive* model of Hodge theory within the framework of Becchi–Rouet–Stora–Tyutin (BRST) formalism where there is the existence of a set of (anti-)co-BRST transformations corresponding to the *usual* nilpotent (anti-)BRST transformations. The *latter* exist in any *arbitrary* dimension of spacetime for the *usual* Stückelberg-modified *massive* Abelian 2-form gauge theory. The modification in the Stückelberg technique is backed by the precise mathematical arguments from the differential geometry where the exterior derivative and Hodge duality operator play the decisive roles. The *modified* version of the Stückelberg technique remains *invariant* under the *discrete* duality transformations which *also* establish a precise and deep connection between the off-shell nilpotent (anti-)BRST and (anti-)co-BRST transformations. We have clarified a simple trick of using the equations of motion to remove the higher derivative terms in the appropriate Lagrangian densities so that our 4D theory can become consistent.

**Keywords:** modified Stückelberg technique; (anti-)BRST symmetries; (anti-)co-BRST symmetries; discrete duality symmetry transformations; fields with negative kinetic terms

**PACS:** 03.70.+k; 11.30.-j; 02.40.-k

## 1. Introduction

The basic concepts and ideas behind the subject of pure mathematics and their applications in the progress of theoretical physics have been intertwined *together* in a meaningful manner since the advent of physics as a specific branch of (the all-encompassing broad field of modern-day) science which incorporates, into its ever-widening folds, other branches, as well. In particular, the recent developments in the domain of theoretical high-energy physics owe a great deal to some of the key ideas and concepts behind pure mathematics. For instance, we know that the concepts of differential geometry have found decisive applications in the realm of theoretical research activities related to the specific topics of gauge theories, gravitational theories, (super)string theories, topological field theories, higher-spin gauge theories, etc. In this context, it is pertinent to point out that the celebrated Stückelberg technique of compensating field(s) [1], responsible for the *massive* field theories (e.g., the Proca theory) to acquire the beautiful gauge symmetry invariance, is also based on the ideas of the differential geometry (see, e.g., [2–5]). In particular, the exterior derivative ($d = dx^\mu \partial_\mu$, $d^2 = 0$) plays a key role (see, e.g., Equation (6) below) in the replacement/modification of the gauge field due to the presence of some compensating field(s) (e.g., a pure scalar field in the context of the Proca theory) which converts the second-class constraints of the original *massive* field theory into the first-class constraints

(see, e.g., [6,7]). The *latter* appear in the expression for the generator of the gauge symmetry transformations (existing in the case of the Stückelberg-modified theory where the mass and gauge symmetry co-exist *together*).

One of the central purposes of our present endeavor is to demonstrate that the *standard* Stückelberg-technique is modified in the context of *massive* Abelian $p$-form ($p = 1, 2, 3, \dots$) gauge theories in $D = 2p$ dimensions of spacetime because such kinds of *massive* theories respect, in addition to the gauge symmetry transformations (that are generated by the first-class constraints in the terminology of Dirac's prescription for the classification of the constraints [6,7]), the dual-gauge symmetry transformations which exist for the gauge-fixed Lagrangian densities of the *above* kinds of theories. In a very recent work [8], we have been able to corroborate the above claim in the context of a 2D Proca (i.e., a *massive* 2D Abelian 1-form) theory. In fact, we have been able to demonstrate that, due to the modified version of the Stückelberg formalism (SF), we obtain the (anti-)BRST and (anti-)co-BRST symmetries for the gauge-fixed Lagrangian density of the Stückelberg-modified 2D Proca theory within the framework of Becchi–Rouet–Stora–Tyutin (BRST) formalism (cf. Appendix A below). It is worthwhile to mention that the *massless* Abelian $p$-form ($p = 1, 2, 3$) gauge theories, in $D = 2p$ dimensions of spacetime, have already been proven to be the field-theoretic models of Hodge theory (see, e.g., [9] and reference therein). Further, we have been able to show that the Stückelberg-modified *massive* 2D Abelian 1-form (i.e., Proca theory) [8] (see, e.g., Appendix A) and 4D massive Abelian 2-form theory (see, e.g., [10]) are, once again, very interesting examples of the *massive* models of Hodge theory within the framework of BRST formalism (see, e.g., [10–12]).

The central theme of our present investigation is to show that the Stückelberg-modified Lagrangian density of the *massive* 4D Abelian 2-form theory respects the (anti-)BRST symmetry transformations in any *arbitrary* dimension of spacetime. However, in the physical four $(3 + 1)$-dimensions of spacetime, *it* respects the (anti-)BRST as well as the (anti-co-)BRST symmetry transformations due to $(i)$ the *modification* in the *standard* Stückelberg technique [cf. Equation (2)] where an axial-vector field $(\tilde{\phi}_\mu)$ *also* appears explicitly (cf. Equation (7)) backed by the precise mathematical arguments, and $(ii)$ the existence of a set of *discrete* duality symmetry transformations under which the *modified* Stückelberg technique (cf. Equation (9)) as well as the 4D Lagrangian density $\mathcal{L}$ (cf. Equation (29)) *both* remain *invariant*. The generalization of *these* discrete *duality* symmetry transformations, within the realm of BRST formalism (cf. Equations (48) and (54)), also establish a precise connection between the (anti-)BRST and (anti-)co-BRST symmetry transformations [9]. We provide proper arguments, however, to demonstrate that the nilpotent (anti-)co-BRST and (anti-)BRST transformations have their own *identities* as they provide the physical realizations [10] of the (co-)exterior derivatives of the differential geometry [2–5], which are also *independent* of each-other.

In our present endeavor, for the sake of brevity, we consider *only* the (co-)BRST invariant Lagrangian density (cf. Section 6) that is the generalization of $\mathcal{L}_{(b_1)}$ (cf. Equation (43)) and establish a *direct* connection between the BRST and co-BRST symmetry transformations due to the existence of a couple of discrete duality symmetry transformations (48) and (54). In an exactly similar fashion, the generalization of the Lagrangian density $\mathcal{L}_{(b_2)}$ (cf. Equation (45)) can be obtained at the *quantum* level (within the framework of BRST formalism) as $\mathcal{L}_{\bar{\mathcal{B}}}$. The *latter* will be anti-BRST as well as anti-co-BRST invariant [10]. Once again, we shall be able to establish the interconnection between the anti-BRST and anti-co-BRST symmetry transformations by exploiting the theoretical potential and power of the discrete *duality* symmetry transformation (48) *plus* (54) at the *quantum* level (see, e.g., [9,10] for details). In addition to this *direct* connection (which is a *novel* observation), there exists another relationship between the co-BRST and BRST transformations (cf. Equation (60)) which provide the physical realization of the relationship between the co-exterior and exterior derivatives of the differential geometry [2–5]. In this context, it is worthwhile to mention that that the discrete duality symmetry transformations (48) *plus* (54) provide the physical realization of the Hodge duality operator of the differential geometry (cf. Equation (60)).

The following key factors have been at the heart of our present investigation. First and foremost, in a very recent work [8], we have discussed the modification of the standard Stückelberg formalism in the context of a *massive* Abelian 1-form (i.e., Proca) theory in two (1 + 1)-dimensions of spacetime. Hence, we have been curious to find its analogue in the context of a *massive* Abelian 2-form theory in the *physical* four (3 + 1)-dimensions of spacetime. Second, we envisage to find out the existence of fields with *negative* kinetic terms on the basis of *symmetry* properties (as has happened in the case of a *modified* 2D Proca theory) because such kinds of *exotic* fields are the possible candidates for dark matter and dark energy, and they play an important role in the context of the cyclic, bouncing, and self-accelerated cosmological models of the Universe (see, e.g., [13–15] and references therein). Third, we desire to establish a *direct* connection between the nilpotent (anti-)BRST and (anti-)co-BRST transformations on the basis of a set of discrete *duality* symmetry transformations (cf. Equations (48) and (54)) *alone* which has *not* been accomplished in our earlier works [9,10]. Fourth, we have developed a simple theoretical trick of using the EL-EoMs to remove the higher derivative terms so that our 4D theory can become renormalizable (cf. Section 3 for details). Fifth, the higher *p*-form ($p = 2, 3, \dots$) gauge theories of *massless* and *massive* varieties are interesting from the point of view of the (super)string theories as they appear in their quantum excitations. Finally, we wish to find the physical realizations of the Hodge duality *operator* of differential geometry [2–5] in terms of the discrete *duality* symmetry transformations within the framework of BRST formalism.

The theoretical material of our present endeavor is organized as follows. In Section 2, we discuss the bare essentials of the gauge symmetry transformations for the standard Stückelberg-modified Lagrangian density in any arbitrary D-dimension of spacetime. Section 3 deals with the modification of the Stückelberg formalism where the exterior derivative and the Hodge duality operator of differential geometry play decisive roles. The subject matter of Section 4 concerns itself with the derivation of the 4D Lagrangian densities that respect the (dual-)gauge symmetry transformations *together* for the gauge-fixed Lagrangian densities, provided *exactly* similar kinds of restrictions are imposed on the (dual-)gauge transformation parameters from the *outside*. Section 5 contains the theoretical discussion on the linearized versions of the gauge-fixed Lagrangian densities and Curci–Ferrari (CF)-type restrictions. In Section 6, we establish a relationship between the BRST and co-BRST symmetry transformations due to the discrete duality symmetry transformations (cf. Equations (48) and (54)) in our BRST-invariant theory. Finally, in Section 7, we make some concluding remarks and point out a few future theoretical directions for further investigation(s).

In Appendix A, we very briefly recapitulate the bare essentials of our earlier work [8] on the Stückelberg-modified 2D Proca theory (where the modified SF has been used). The theoretical content of Appendix B is devoted to the generalization of the *classical* symmetry transformations (37) and (35) to their *quantum* counterparts and (co-)BRST symmetry transformations for the *appropriate* (co-)BRST invariant Lagrangian density. It turns out, however, that the Lagrangian density (49) is appropriate and unique as it satisfies all the essential requirements of a *properly* gauge-fixed and (anti-)BRST invariant theory.

*Convention and Notations*: We follow the convention of the left-derivative w.r.t. *all* the fermionic (i.e., $\bar{C}_\mu$, $C_\mu$, $\bar{C}$, $C$, $\rho$, $\lambda$,) fields of our theory in the context of the derivation of the equations of motions, definition of the conjugate momenta, deduction of the Noether conserved currents and charges, etc. The 4D Levi–Civita tensor is denoted by $\varepsilon_{\mu\nu\lambda\xi}$ with conventions: $\varepsilon_{0123} = +1 = -\varepsilon^{0123}$ and $\varepsilon_{\mu\nu\lambda\xi}\,\varepsilon^{\mu\nu\lambda\xi} = -4!$, $\varepsilon_{\mu\nu\lambda\xi}\,\varepsilon^{\mu\nu\lambda\rho} = -3!\,\delta^\rho_\xi$, $\varepsilon_{\mu\nu\lambda\xi}\,\varepsilon^{\mu\nu\rho\sigma} = -2!\,(\delta^\rho_\lambda\,\delta^\sigma_\xi - \delta^\rho_\xi\,\delta^\sigma_\lambda)$, etc., where the Greek indices $\mu$, $\nu$, $\lambda$, $\dots = 0, 1, 2, 3$ stand for the time and space directions, and the Latin indices $i, j, k \dots = 1, 2, 3$ correspond to the space directions *only*. Hence, the 3D Levi–Civita tensor is $\epsilon_{ijk} = \varepsilon_{0ijk}$. The background *flat* 4D Minkowskian spacetime manifold is endowed with a *flat* metric tensor $\eta_{\mu\nu} = \text{diag}\,(+1, -1, -1, -1)$ so that the dot product between two *non-null* 4-vectors $P_\mu$ and $Q_\mu$ is represented by: $P \cdot Q = \eta_{\mu\nu}\,P^\mu\,Q^\nu = P_0\,Q_0 - P_i\,Q_i$. We denote the nilpotent

(anti-)BRST symmetry transformations by $s_{(a)b}$, and the notations $s_{(a)d}$ stand for the nilpotent (anti-) dual (i.e., (anti-)co-BRST symmetry transformations.

*Standard Definition*: On a compact manifold without a boundary, we have a set of *three* operators ($d$, $\delta$, $\Delta$) which are known as the de Rham cohomological operators of differential geometry. The operators ($\delta$) $d$ are called the (co-)exterior derivatives that are connected with each other by the algebraic relationship: $\delta = \pm * d *$ where $*$ is the Hodge duality operator on the above manifold. These operators satisfy the algebra: $d^2 = \delta^2 = 0$, $\Delta = (d + \delta)^2 = \{d, \delta\}$, $[\Delta, d] = [\Delta, \delta] = 0$, where $\Delta$ is the Laplacian operator [2–5]. This algebra (which is *not* a Lie algebra) is popularly known as the Hodge algebra and $\Delta$ behaves similar to a Casimir operator for the whole algebra (but *not* in the Lie algebraic sense). We shall be frequently using the names of these cohomological operators (see, e.g., [2–5]) of differential geometry in our present endeavor in appropriate places.

## 2. Preliminaries: Stückelberg Formalism in Any Arbitrary Dimension of Spacetime

We begin with any arbitrary D-dimensional Lagrangian density ($\mathcal{L}_0$) for the *massive* Abelian 2-form ($B^{(2)} = [(d\,x^\mu \wedge d\,x^\nu)/2!]\,B_{\mu\nu}$) theory with the anti-symmetric tensor ($B_{\mu\nu} = -B_{\nu\mu}$) field (carrying the rest mass $m$) as follows (see, e.g., [16] and references therein)

$$\mathcal{L}_0 = \frac{1}{12} H^{\mu\nu\lambda} H_{\mu\nu\lambda} - \frac{m^2}{4} B_{\mu\nu} B^{\mu\nu}, \tag{1}$$

where $H^{(3)} = d\,B^{(2)} \equiv [(d\,x^\mu \wedge d\,x^\nu \wedge d\,x^\lambda)/3!]\,H_{\mu\nu\lambda}$ defines the kinetic term (with the field-strength tensor $H_{\mu\nu\lambda} = \partial_\mu B_{\nu\lambda} + \partial_\nu B_{\lambda\mu} + \partial_\lambda B_{\mu\nu}$ for the anti-symmetric tensor field $B_{\mu\nu}$ where the Greek indices $\mu, \nu, \lambda \ldots = 0, 1, \ldots, D-1$). It is straightforward to check that the Euler–Lagrange (EL) equation of motion (EoM): $\partial_\mu H^{\mu\nu\lambda} + m^2 B^{\nu\lambda} = 0$ implies the subsidiary conditions: $\partial_\nu B^{\nu\lambda} = \partial_\lambda B^{\nu\lambda} = 0$, which emerge out from *it* for $m^2 \neq 0$. As a consequence, we observe that $B_{\mu\nu}$ field obeys the Klein–Gordon equation $(\Box + m^2)\,B_{\mu\nu} = 0$ with a definite rest mass $m$. We note that the *massive* Lagrangian density (1) does *not* respect the gauge transformation due to the fact that it is endowed with the second-class constraints in the terminology of Dirac's prescription for the classification scheme of constraints (*because the gauge symmetries are generated by the first-class constraints [6,7]*).

The gauge symmetry transformations can be restored for the *modified* version of the standard Lagrangian density (1) if we exploit the basic theoretical methodology of the Stückelberg formalism (SF) related to the compensating field(s). In other words, due to SF, we replace the *basic* antisymmetric Abelian 2-form field $B_{\mu\nu}$ as follows [16,17]:

$$B_{\mu\nu} \longrightarrow B_{\mu\nu} \mp \frac{1}{m} (\partial_\mu \phi_\nu - \partial_\nu \phi_\mu), \tag{2}$$

where the Abelian 1-form $\Phi^{(1)} = d\,x^\mu \phi_\mu$ defines the *vector* field $\phi_\mu$. It is straightforward to check that $H_{\mu\nu\lambda} = \partial_\mu B_{\nu\lambda} + \partial_\nu B_{\lambda\mu} + \partial_\lambda B_{\mu\nu}$ remains invariant under (2). We note that the mass dimension of $B_{\mu\nu}$ and $\phi_\mu$ fields are the *same* in the D-dimensional Minkowskian flat spacetime when we use the natural units: $\hbar = c = 1$. Hence, the rest mass $m$ should be present in the denominator of Equation (2) to balance the mass dimension on the l.h.s and r.h.s of Equation (2). The *mass* term in Equation (1) changes as follows, due to (2), namely;

$$-\frac{m^2}{4} B_{\mu\nu} B^{\mu\nu} \longrightarrow -\frac{m^2}{4} \left[B_{\mu\nu} \mp \frac{1}{m}(\partial_\mu \phi_\nu - \partial_\nu \phi_\mu)\right] \left[B^{\mu\nu} \mp \frac{1}{m}(\partial^\mu \phi^\nu - \partial^\nu \phi^\mu)\right]. \tag{3}$$

Let us define an Abelian 2-form $F^{(2)} = d\Phi^{(1)} = [(dx^\mu \wedge dx^\nu)/2!]\,\Phi_{\mu\nu}$ where $\Phi_{\mu\nu} = \partial_\mu\phi_\nu - \partial_\nu\phi_\mu$ is the antisymmetric field strength tensor for the vector field $\phi_\mu$. With all *these* inputs, we obtain the Stückelberg-modified Lagrangian density $\mathcal{L}_S$ from $\mathcal{L}_0$ as

$$\mathcal{L}_0 \longrightarrow \mathcal{L}_S = \frac{1}{12} H^{\mu\nu\lambda} H_{\mu\nu\lambda} - \frac{m^2}{4} B_{\mu\nu} B^{\mu\nu} \pm \frac{m}{2} B_{\mu\nu} \Phi^{\mu\nu} - \frac{1}{4} \Phi_{\mu\nu} \Phi^{\mu\nu}, \tag{4}$$

which respects the following local, continuous and infinitesimal *classical* gauge symmetry transformations $\delta_g$, namely

$$\begin{aligned}
\delta_g H_{\mu\nu\lambda} &= 0, & \delta_g \phi_\mu &= \pm(\partial_\mu\Lambda - m\Lambda_\mu), \\
\delta_g B_{\mu\nu} &= -(\partial_\mu\Lambda_\nu - \partial_\nu\Lambda_\mu), & \delta_g \Phi_{\mu\nu} &= \mp m(\partial_\mu\Lambda_\nu - \partial_\nu\Lambda_\mu),
\end{aligned} \tag{5}$$

where the Lorentz scalar $\Lambda(x)$ and Lorentz vector $\Lambda_\mu(x)$ are the infinitesimal *local* gauge symmetry transformation parameters. It is important to point out that there is a stage-one reducibility in the theory because the transformation $\phi_\mu \to \phi_\mu \pm \partial_\mu\Lambda$ can be accommodated in the standard Stückelberg technique (considered in Equation (2)) *without* changing *it* in any way. This is why, in the gauge transformation of $\phi_\mu$ field (cf. Equation (5)), we have the local Lorentz *scalar* transformation parameter $\Lambda(x)$. It is straightforward to check that $\delta_g \mathcal{L}_S = 0$, implying that the Stückelberg-modified Lagrangian density $\mathcal{L}_S$ respects the infinitesimal and continuous *local* gauge symmetry transformations (5) in a *perfect* manner. We mention, in passing, that the second-class constraints of $\mathcal{L}_0$ have been converted into the first-class constraints (due to the introduction of the Stückelberg polar vector field $\phi_\mu$ in (2)). The ensuing first-class constraints are the generator for the infinitesimal gauge symmetry transformations $(\delta_g)$ in (5). These statements are *true* for our theory in any arbitrary D-dimension of Minkowskian flat spacetime [17].

## 3. Massive 4D Abelian 2-Form Theory: Modified SF

In the differential form terminology, the standard Stückelbergtechnique (2), defined for any arbitrary D-dimension of spacetime, can be re-expressed as follows [17]:

$$B^{(2)} \longrightarrow B^{(2)} \mp \frac{1}{m} F^{(2)} \equiv B^{(2)} \mp \frac{1}{m} d\Phi^{(1)}. \tag{6}$$

This also establishes the invariance of $H^{(3)} = dB^{(2)}$ under (2) because of the nilpotency ($d^2 = 0$) of the exterior derivative ($d$). In the physical four (3 + 1)-dimensional (4D) *flat* spacetime, the theoretical technique (6) of the *standard* Stückelberg formalism can be *modified* in the following manner (in the language of differential forms), namely;

$$B^{(2)} \longrightarrow B^{(2)} \mp \frac{1}{m} d\Phi^{(1)} \mp \frac{1}{m} * d\tilde{\Phi}^{(1)}, \tag{7}$$

where the first *two* terms of the r.h.s. have already been explained. In the *third* term, on the r.h.s., we have taken the axial-vector 1-form $\tilde{\Phi}^{(1)} = dx^\mu \tilde{\phi}_\mu$ with the axial-vector field $\tilde{\phi}_\mu$. A pseudo 2-form $\tilde{F}^{(2)} = d\tilde{\Phi}^{(1)} = [(dx^\mu \wedge dx^\nu)/2!]\,\tilde{\Phi}_{\mu\nu}$ has been constructed from $\tilde{\Phi}^{(1)}$ by applying an exterior derivative on it so that we obtain $\tilde{\Phi}_{\mu\nu} = \partial_\mu\tilde{\phi}_\nu - \partial_\nu\tilde{\phi}_\mu$. To bring the parity of $B^{(2)}$, $F^{(2)} = d\Phi^{(1)}$ and the *pseudo* 2-form $\tilde{F}^{(2)}$ on *equal* footing[1], it is essential to obtain an *ordinary* 2-form *from* the *pseudo* 2-form $\tilde{F}^{(2)}$ by operating a *single* Hodge duality operator $*$ on it. This mathematical technique (on the 4D spacetime manifold) leads to

$$* \tilde{F}^{(2)} = *\left(\frac{dx^\mu \wedge dx^\nu}{2!}\right)\tilde{\Phi}_{\mu\nu} \equiv \frac{1}{2!}(dx^\mu \wedge dx^\nu)\,\tilde{f}_{\mu\nu}, \tag{8}$$

where $\tilde{f}_{\mu\nu} = \frac{1}{2}\,\varepsilon_{\mu\nu\lambda\xi}\,\tilde{\Phi}^{\lambda\xi} \equiv \varepsilon_{\mu\nu\lambda\xi}\,\partial^\lambda\,\tilde{\phi}^\xi$. Thus, in the language of a set of antisymmetric tensors $(B_{\mu\nu},\,\Phi_{\mu\nu},\,\tilde{f}_{\mu\nu})$, we have obtained the following from the *modified* version of the 4D Stückelberg technique (7), namely;

$$
\begin{aligned}
B_{\mu\nu} \quad\longrightarrow\quad & B_{\mu\nu} \mp \frac{1}{m}\,(\partial_\mu\,\phi_\nu - \partial_\nu\,\phi_\mu) \mp \frac{1}{m}\,\varepsilon_{\mu\nu\lambda\xi}\,\partial^\lambda\,\tilde{\phi}^\xi \\
\equiv\quad & B_{\mu\nu} \mp \frac{1}{m}\,\Phi_{\mu\nu} \mp \frac{1}{m}\,\tilde{f}_{\mu\nu} \equiv B_{\mu\nu} \mp \frac{1}{m}\left(\Phi_{\mu\nu} + \frac{1}{2}\,\varepsilon_{\mu\nu\lambda\xi}\,\tilde{\Phi}^{\lambda\xi}\right).
\end{aligned} \tag{9}
$$

It is very interesting to state that the above *modified* 4D Stückelberg technique remains form-invariant under the following discrete *duality* symmetry transformations:

$$
B_{\mu\nu} \to \mp i\,\tilde{B}_{\mu\nu} \equiv \mp\frac{i}{2!}\,\varepsilon_{\mu\nu\lambda\xi}\,B^{\lambda\xi}, \qquad \phi_\mu \to \pm i\,\tilde{\phi}_\mu, \qquad \tilde{\phi}_\mu \to \mp i\,\phi_\mu, \tag{10}
$$

where $\tilde{B}_{\mu\nu} = \frac{1}{2!}\,\varepsilon_{\mu\nu\lambda\xi}\,B^{\lambda\xi}$ emerges out from the self-duality condition: $* B^{(2)} = [(d\,x^\mu \wedge d\,x^\nu)/2!]\,B_{\mu\nu}$, which leads to the derivation of the *dual* Abelian 2-form (in 4D) as follows:

$$
\tilde{B}^{(2)} = \left(\frac{d\,x^\mu \wedge d\,x^\nu}{2!}\right)\left[\frac{1}{2!}\,\varepsilon_{\mu\nu\lambda\xi}\,B^{\lambda\xi}\right] \equiv \left(\frac{d\,x^\mu \wedge d\,x^\nu}{2!}\right)\tilde{B}_{\mu\nu}. \tag{11}
$$

We shall see that the *discrete* duality symmetry transformations in (10) will play a very important role, later on, as *its* generalization (within the framework of BRST formalism) will provide the analogue of the Hodge duality $*$ operator of differential geometry. We would like to lay emphasis on the fact that the root cause behind the existence of the discrete *duality* symmetry transformations in (10) is the *self-duality* condition on the Abelian 2-form $(B^{(2)} = [(d\,x^\mu \wedge d\,x^\nu)/2!]\,B_{\mu\nu})$ in the physical four $(3+1)$-dimensions of spacetime.

It is an elementary exercise to note that the *mass* term of Equation (1) transforms, under the *modified* Stückelberg technique (cf. Equation (9)), as

$$
\begin{aligned}
-\frac{m^2}{4}\,B_{\mu\nu}\,B^{\mu\nu} \longrightarrow\quad & -\quad \frac{m^2}{4}\,B_{\mu\nu}\,B^{\mu\nu} \pm \frac{m}{2}\,B_{\mu\nu}\left(\Phi^{\mu\nu} + \frac{1}{2}\varepsilon^{\mu\nu\rho\sigma}\,\tilde{\Phi}_{\rho\sigma}\right) \\
& -\quad \frac{1}{4}\,\Phi_{\mu\nu}\,\Phi^{\mu\nu} + \frac{1}{4}\tilde{\Phi}_{\mu\nu}\,\tilde{\Phi}^{\mu\nu},
\end{aligned} \tag{12}
$$

modulo a total spacetime derivative $\partial_\mu\left[-\varepsilon^{\mu\nu\lambda\xi}\,\phi_\nu\,\partial_\lambda\,\tilde{\phi}_\xi\right]$ which emerges out from a term $(-\frac{1}{2}\,\tilde{f}_{\mu\nu}\,\Phi^{\mu\nu})$ that appears in Equation (12) due to the substitution (9) for the modified version of the antisymmetric field $B_{\mu\nu}$. We also point out that the kinetic term $((1/12)\,H_{\mu\nu\lambda}\,H^{\mu\nu\lambda})$ *also* transforms under (9) because it is straightforward to note that we have the following

$$
\begin{aligned}
H_{\mu\nu\lambda} \longrightarrow\quad & H_{\mu\nu\lambda} \mp \frac{1}{m}\,(\partial_\mu\,\Phi_{\nu\lambda} + \partial_\nu\,\Phi_{\lambda\mu} + \partial_\lambda\,\Phi_{\mu\nu}) \\
& \mp \frac{1}{m}\,(\partial_\mu\,\tilde{f}_{\nu\lambda} + \partial_\nu\,\tilde{f}_{\lambda\mu} + \partial_\lambda\,\tilde{f}_{\mu\nu}),
\end{aligned} \tag{13}
$$

where $\Phi_{\mu\nu} = \partial_\mu\,\phi_\nu - \partial_\nu\,\phi_\mu$ and $\tilde{f}_{\mu\nu} = \varepsilon_{\mu\nu\lambda\xi}\,\partial^\lambda\,\tilde{\phi}^\xi$. We note that the *second* term on the r.h.s. of the above equation turns out to be *zero*. However, the *third* term exists as:

$$
\begin{aligned}
\mp\frac{1}{m}\,\Sigma_{\mu\nu\lambda} \quad=\quad & \mp\frac{1}{m}\,(\partial_\mu\,\tilde{f}_{\nu\lambda} + \partial_\nu\,\tilde{f}_{\lambda\mu} + \partial_\lambda\,\tilde{f}_{\mu\nu}) \\
\equiv\quad & \mp\frac{1}{m}\,(\varepsilon_{\mu\nu\rho\sigma}\,\partial_\lambda + \varepsilon_{\nu\lambda\rho\sigma}\,\partial_\mu + \varepsilon_{\lambda\mu\rho\sigma}\,\partial_\nu)\,(\partial^\rho\,\tilde{\phi}^\sigma).
\end{aligned} \tag{14}
$$

Thus, we have to find the *exact* value of the following (for the changes in the kinetic term due to the modified version of 4D Stückelberg technique (cf. Equation (9))), namely;

$$
\frac{1}{12}\,H_{\mu\nu\lambda}\,H^{\mu\nu\lambda} \implies \frac{1}{12}\,H_{\mu\nu\lambda}\,H^{\mu\nu\lambda} \mp \frac{1}{6\,m}\,H_{\mu\nu\lambda}\,\Sigma^{\mu\nu\lambda} + \frac{1}{12\,m^2}\,\Sigma_{\mu\nu\lambda}\,\Sigma^{\mu\nu\lambda}, \tag{15}
$$

where we have taken into account $H_{\mu\nu\lambda} \longrightarrow H_{\mu\nu\lambda} \mp \frac{1}{m} \Sigma_{\mu\nu\lambda}$ (cf. Equations (13) and (14)). We focus on the *second* term $[\mp(1/6\,m)\,H^{\mu\nu\lambda}\,\Sigma_{\mu\nu\lambda}]$, which can be explicitly written as:

$$\mp \frac{1}{6\,m}\,H^{\mu\nu\lambda}\,\left[\varepsilon_{\mu\nu\rho\sigma}\,\partial_\lambda + \varepsilon_{\nu\lambda\rho\sigma}\,\partial_\mu + \varepsilon_{\lambda\mu\rho\sigma}\,\partial_\nu\right](\partial^\rho\tilde{\phi}^\sigma). \tag{16}$$

The *first* term on the r.h.s. of the above equation contributes the following (modulo a total spacetime derivative), namely;

$$\pm \frac{1}{6\,m}\,(\partial_\lambda\,H^{\lambda\mu\nu})\,\varepsilon_{\mu\nu\rho\sigma}\,(\partial^\rho\tilde{\phi}^\sigma). \tag{17}$$

It is self-evident that there are *three* derivatives in the above expression because $H_{\mu\nu\lambda}$ contains *one* derivative. Thus, the expression in (17) belongs to a *higher* derivative term for our 4D theory. It is worthwhile to mention here that in our earlier work [10], the higher derivative terms have been ignored. This is why relevant terms in the Lagrangian density have been obtained by the *trial* and *error* method. However, we note, in our present endeavor, that one can remove a *single* derivative by using the on-shell condition: $\partial_\lambda\,H^{\lambda\mu\nu} + m^2\,B^{\mu\nu} = 0$, which is equivalent to the EL-EoM: $(\Box + m^2)\,B_{\mu\nu} = 0$. It is due to the presence of the higher derivative term that the substitution of the on-shell condition does *not* make this term (in the Lagrangian density) equal to zero. This should be contrasted *against* the use of the on-shell conditions in the context of the simple cases of (i) the Dirac Lagrangian density and (ii) the pure (Klein–Gorden) scalar field Lagrangian density (where there are *no* higher order derivatives). All the *three* terms, on the r.h.s. of (16), *individually* contribute to the *same* result, which can be added *together* to yield

$$\mp \frac{m}{2}\,\varepsilon_{\mu\nu\lambda\xi}\,B^{\mu\nu}\,(\partial^\lambda\,\tilde{\phi}^\xi) \equiv \mp \frac{m}{4}\,\varepsilon^{\mu\nu\lambda\xi}\,B_{\mu\nu}\,\tilde{\Phi}_{\lambda\xi}, \tag{18}$$

where $\tilde{\Phi}_{\lambda\xi} = \partial_\lambda\,\tilde{\phi}_\xi - \partial_\xi\,\tilde{\phi}_\lambda$. It is interesting to point out that the above term has been incorporated into the BRST invariant Lagrangian density of our earlier work [10] on the *basis* of the trial and error method. However, as is self-evident, we have derived this term correctly in our present endeavor, which is motivated by our earlier work on the Stückelberg-modified 2D Proca theory [8], where we have exploited a *similar* kind of trick to remove the higher derivative terms. The mass term in Equation (18) is similar to the topological mass term of the $B \wedge F$ theory. In the *latter* theory, the 4D Abelian 2-form theory *also* incorporates the Maxwell Abelian 1-form ($A^{(1)} = d\,x^\mu\,A_\mu$) gauge field with the curvature 2-form $F^{(2)} = d\,A^{(1)}$. There are many ways to derive (18) from (16). However, we have chosen one of the *simplest* methods to derive Equation (18), which is *not* a *higher* derivative *culprit* term for our 4D Abelian 2-form massive theory.

We now focus on the *exact* and *explicit* computation of the *third* term on the r.h.s. of (15). It is evident that, for a 4D Abelian 2-form theory, this *third* term is a higher derivative term because it contains *four* derivatives in it. A close look at (14) shows that there will be a *total* of nine terms when we take into account $((1/12\,m^2)\,\Sigma_{\mu\nu\lambda}\,\Sigma^{\mu\nu\lambda})$ and write the expression for $\Sigma_{\mu\nu\lambda}$ from Equation (16). However, it turns out that only *three* of them contribute to the Lagrangian density, and the rest of the *six* terms are found to be *total* spacetime derivatives. Hence, they can be ignored as the dynamics of the theory does not depend on them. Let us focus on the *first* existing term, which is equal to the following:

$$\frac{1}{12\,m^2}\,\varepsilon^{\mu\nu\rho\sigma}\,\partial^\lambda\,(\partial_\rho\,\tilde{\phi}_\sigma)\,\varepsilon_{\mu\nu\alpha\beta}\,\partial_\lambda\,(\partial^\alpha\,\tilde{\phi}^\beta)$$

$$= -\frac{1}{6\,m^2}\,[\partial^\lambda\,(\partial_\alpha\,\tilde{\phi}_\beta - \partial_\beta\,\tilde{\phi}_\alpha)]\,[\partial_\lambda\,(\partial^\alpha\,\tilde{\phi}^\beta)]$$

$$\equiv \frac{1}{6\,m^2}\,\partial^\lambda\,(\partial^\alpha\,\tilde{\Phi}_{\alpha\beta})\,(\partial_\lambda\,\tilde{\phi}^\beta), \tag{19}$$

where we have dropped a total spacetime derivative term, as it will *not* affect the dynamics At this stage, to remove the derivatives, we use the EoM: $\partial_\alpha \tilde{\Phi}^{\alpha\beta} + m^2 \tilde{\phi}^\beta = 0$. Thus, the *final* expression on the r.h.s. of Equation (19) is

$$-\frac{1}{6}\,(\partial^\lambda\,\tilde{\phi}_\beta)\,(\partial_\lambda\,\tilde{\phi}^\beta) \equiv \frac{1}{6}\,\tilde{\phi}_\beta\,\Box\,\tilde{\phi}^\beta, \tag{20}$$

where we have dropped a total spacetime derivative term. Using the Klein–Gordon equation[2]: $(\Box + m^2)\,\tilde{\phi}_\mu = 0$, we obtain the following (from the *first* contribution), namely;

$$\frac{1}{12\,m^2}\,\varepsilon^{\mu\nu\rho\sigma}\,\partial^\lambda\,(\partial_\rho\,\tilde{\phi}_\sigma)\,\varepsilon_{\mu\nu\alpha\beta}\,\partial_\lambda\,(\partial^\alpha\,\tilde{\phi}^\beta) \equiv -\frac{m^2}{6}\,\tilde{\phi}_\beta\,\tilde{\phi}^\beta. \tag{21}$$

It is obvious that there are *three* such contributions in the *total* evaluation of the *third* term $((1/12\,m^2)\,\Sigma_{\mu\nu\lambda}\,\Sigma^{\mu\nu\lambda})$ on the r.h.s. of Equation (15). Thus, ultimately, we obtain the following *exact* and *explicit* expression from all *three* existing terms, namely;

$$\frac{1}{12\,m^2}\,\Sigma_{\mu\nu\lambda}\,\Sigma^{\mu\nu\lambda} = -\frac{m^2}{2}\,\tilde{\phi}_\mu\,\tilde{\phi}^\mu. \tag{22}$$

Going from Equation (20) to Equation (22) is essential and interesting for our purpose. It is clear that the above term is *not* a *culprit* term, and it is useful to us for our further discussions. The *total* terms on the r.h.s. of Equation (15) can be re-expressed as follows:

$$\frac{1}{12}\,H^{\mu\nu\lambda}\,H_{\mu\nu\lambda} \mp \frac{m}{2}\,\varepsilon_{\mu\nu\lambda\xi}\,B^{\mu\nu}\,(\partial^\lambda\,\tilde{\phi}^\xi) - \frac{m^2}{2}\,\tilde{\phi}_\mu\,\tilde{\phi}^\mu$$
$$\equiv -\frac{1}{8}\,\varepsilon^{\mu\nu\lambda\xi}\,(\partial_\nu\,B_{\lambda\xi})\,\varepsilon_{\mu\alpha\beta\gamma}\,(\partial^\alpha\,B^{\beta\gamma})$$
$$\pm \frac{m}{2}\,\varepsilon_{\mu\nu\lambda\xi}\,(\partial^\mu\,B^{\nu\lambda})\,\tilde{\phi}^\xi - \frac{m^2}{2}\,\tilde{\phi}_\mu\,\tilde{\phi}^\mu, \tag{23}$$

where we have dropped a total spacetime derivative term and used the following exact equality, namely;

$$\frac{1}{12}\,H^{\mu\nu\lambda}\,H_{\mu\nu\lambda} = -\frac{1}{8}\,\varepsilon^{\mu\nu\lambda\xi}\,(\partial_\nu\,B_{\lambda\xi})\,\varepsilon_{\mu\alpha\beta\gamma}\,(\partial^\alpha\,B^{\beta\gamma}). \tag{24}$$

The correctness of the above equality can be checked explicitly by using the well-known property of the 4D Levi–Civita tensor, where one index (i.e., $\mu$) is contracted. It is straightforward to observe that the *final* expression for (23) can be written as

$$-\frac{1}{2}\left[\frac{1}{4}\,\varepsilon^{\mu\nu\lambda\xi}\,(\partial_\nu\,B_{\lambda\xi})\,\varepsilon_{\mu\alpha\beta\gamma}\,(\partial^\alpha\,B^{\beta\gamma}) \pm m\,\tilde{\phi}_\mu\,\varepsilon^{\mu\nu\lambda\xi}\,(\partial_\nu\,B_{\lambda\xi}) + m^2\,\tilde{\phi}_\mu\,\tilde{\phi}^\mu\right], \tag{25}$$

where we have used the following

$$\pm\frac{m}{2}\,\varepsilon_{\mu\nu\lambda\xi}\,(\partial^\mu\,B^{\nu\lambda})\,\tilde{\phi}^\xi = \mp\frac{m}{2}\,\tilde{\phi}_\mu\,\varepsilon^{\mu\nu\lambda\xi}\,(\partial_\nu\,B_{\lambda\xi}), \tag{26}$$

to express (15) (and/or (23) and/or (25)) as a squared term, namely;

$$\frac{1}{12}\,H_{\mu\nu\lambda}\,H^{\mu\nu\lambda} \mp \frac{1}{6\,m}\,H_{\mu\nu\lambda}\,\Sigma^{\mu\nu\lambda} + \frac{1}{12\,m^2}\,\Sigma_{\mu\nu\lambda}\,\Sigma^{\mu\nu\lambda}$$
$$= -\frac{1}{2}\left[\frac{1}{2}\,\varepsilon_{\mu\nu\lambda\xi}\partial^\nu\,B^{\lambda\xi} \pm m\,\tilde{\phi}_\mu\right]^2, \tag{27}$$

which is nothing but the explicit expression for Equation (25). Thus, we note that the *final* version of the Lagrangian density (with the modified SF (cf. Equation (9))) is as follows[3]

$$
\begin{aligned}
\mathcal{L}_S^{(m)} &= -\frac{1}{2}\left[\frac{1}{2}\varepsilon_{\mu\nu\lambda\xi}\partial^\nu B^{\lambda\xi} \pm m\,\tilde{\phi}_\mu\right]^2 - \frac{m^2}{4}\,B_{\mu\nu}\,B^{\mu\nu} \\
&\quad \pm \frac{m}{2}\left[\Phi^{\mu\nu} + \frac{1}{2}\varepsilon^{\mu\nu\rho\sigma}\,\tilde{\Phi}_{\rho\sigma}\right]B_{\mu\nu} - \frac{1}{4}\,\Phi_{\mu\nu}\,\Phi^{\mu\nu} + \frac{1}{4}\tilde{\Phi}_{\mu\nu}\,\tilde{\Phi}^{\mu\nu},
\end{aligned}
\tag{28}
$$

where we have taken the inputs from Equations (12) and (27), and the superscript $(m)$ on *this* Lagrangian density denotes that we have taken into account the help of the *modified* SF (cf. Equation (9)) in the modified version of the Lagrangian density (28).

We end this section with the *final* remark that we can add a gauge-fixing term for the Abelian 2-form field $(B_{\mu\nu})$, the axial-vector field $(\tilde{\phi}_\mu)$, and the polar vector field $(\phi_\mu)$ so that we can quantize the theory (described by the Lagrangian density (28)). At this stage, the role of the co-exterior derivative $(\delta = \pm * d *, \delta^2 = 0)$ becomes quite essential as we note that $\delta B^{(2)} = (\partial^\nu B_{\nu\mu})\,d\,x^\mu$, $\delta\,\Phi^{(1)} = (\partial\cdot\phi)$, $\delta\tilde{\Phi}^{(1)} = (\partial\cdot\tilde{\phi})$ where $\delta = -*d*$ is the co-exterior derivative defined on the 4D spacetime (which is an *even* dimensional Minkowskian spacetime manifold). The full Lagrangian density, with the gauge-fixing terms, is

$$
\begin{aligned}
\mathcal{L} = \mathcal{L}_S^{(m)} + \mathcal{L}_{gf} &\equiv -\frac{1}{2}\left[\frac{1}{2}\varepsilon_{\mu\nu\lambda\xi}\partial^\nu B^{\lambda\xi} \pm m\,\tilde{\phi}_\mu\right]^2 - \frac{m^2}{4}\,B_{\mu\nu}\,B^{\mu\nu} \\
&\quad \pm \frac{m}{2}\left[\Phi^{\mu\nu} + \frac{1}{2}\varepsilon^{\mu\nu\rho\sigma}\,\tilde{\Phi}_{\rho\sigma}\right]B_{\mu\nu} - \frac{1}{4}\,\Phi_{\mu\nu}\,\Phi^{\mu\nu} + \frac{1}{4}\tilde{\Phi}_{\mu\nu}\,\tilde{\Phi}^{\mu\nu} \\
&\quad + \frac{1}{2}\left[\partial^\nu B_{\nu\mu} \pm m\,\phi_\mu\right]^2 + \frac{1}{2}\,(\partial\cdot\tilde{\phi})^2 - \frac{1}{2}\,(\partial\cdot\phi)^2,
\end{aligned}
\tag{29}
$$

where the gauge-fixing term $(\frac{1}{2}\,(\partial^\nu B_{\nu\mu} \pm m\,\phi_\mu)^2)$ is similar to the t'Hooft gauge in the context of the Stückelberg-*modified* Proca theory [8,17]. We point out that the above gauge-fixed Lagrangian density respects the *duality* symmetry transformations (10). The *latter*, it goes without saying, are also respected by the modified SF that has been defined in Equation (9). The equations of motion, satisfied by the basic fields $(B_{\mu\nu}, \phi_\mu, \tilde{\phi}_\mu)$, are the Klein–Gordon equations: $(\Box + m^2)\,B_{\mu\nu} = 0$, $(\Box + m^2)\,\phi_\mu = 0$, $(\Box + m^2)\,\tilde{\phi}_\mu = 0$, which emerge out from the Lagrangian density (29). This observation establishes that (i) *all* the fields have the rest mass $m$ and our gauge-fixing procedure is *correct*, where the fields $(\phi_\mu, \tilde{\phi}_\mu)$ have been incorporated into (29) on the basis of the consideration of the proper *mass* dimension in 4D, and (ii) the EL-EoMs used, in this section, to remove the higher derivative terms are *not* far-fetched.

## 4. Final Forms of the Gauge-Fixed Lagrangian Densities: Massive Free 4D Abelian 2-Form Theory

The gauge-fixing term and the kinetic term that have been obtained in (29) can be further generalized. It is a textbook[4] material that one can incorporate a pure scalar field $(\phi)$ with mass dimension one (i.e., $[M]$) in the *massless* case of Abelian 2-form gauge theory in the following explicit manner:

$$
B^\mu\,(\partial^\nu B_{\nu\mu} - \partial_\mu\,\phi) - \frac{1}{2}\,B^\mu\,B_\mu \equiv \frac{1}{2}\,(\partial^\nu B_{\nu\mu} - \partial_\mu\,\phi)^2,
\tag{30}
$$

where, in the 4D Minkowskian spacetime, the mass dimensions of the Nakanishi–Lautrup type auxiliary field $B^\mu$ and $(\partial^\nu B_{\nu\mu} - \partial_\mu\,\phi)$ are two (i.e., $[M]^2$) in the natural units $(\hbar = c = 1)$. In the case of our *massive* 4D Abelian 2-form theory, we can choose the analogue of (30) by generalizing the gauge-fixing term of (29). However, since our theory is duality-invariant, we have to generalize the kinetic term, too, by incorporating a pseudo-scalar field $(\tilde{\phi})$ with a mass dimension of one (i.e., $[M]$). To be consistent with the Curci-Ferrari restriction that has been derived by the superfield approach to BRST formalism in the context of Abelian

2-form theory (see, e.g., [18] for details), we choose the pure scalar and pseudo-scalar fields with a factor of $(\pm 1/2)$ to begin with. However, only one sign will be taken into consideration for a specific Lagrangian density of our theory (later on).

As a consequence of the above arguments, we have the following modifications of the gauge-fixed Lagrangian density (29), namely;

$$
\begin{aligned}
\mathcal{L} \longrightarrow \mathcal{L}_{(1)} = & -\frac{1}{2}\left[\frac{1}{2}\varepsilon_{\mu\nu\lambda\xi}\partial^\nu B^{\lambda\xi} \pm m\,\tilde{\phi}_\mu \mp \frac{1}{2}\partial_\mu\tilde{\phi}\right]^2 - \frac{m^2}{4}\,B_{\mu\nu}\,B^{\mu\nu} \\
& \pm \frac{m}{2}\left[\Phi^{\mu\nu} + \frac{1}{2}\varepsilon^{\mu\nu\rho\sigma}\tilde{\Phi}_{\rho\sigma}\right]B_{\mu\nu} - \frac{1}{4}\,\Phi_{\mu\nu}\,\Phi^{\mu\nu} + \frac{1}{4}\tilde{\Phi}_{\mu\nu}\,\tilde{\Phi}^{\mu\nu} \\
& + \frac{1}{2}\left[\partial^\nu B_{\nu\mu} \pm m\,\phi_\mu \mp \frac{1}{2}\partial_\mu\phi\right]^2 + \frac{1}{2}\left(\partial\cdot\tilde{\phi} + \frac{m}{2}\,\tilde{\phi}\right)^2 \\
& - \frac{1}{2}\left(\partial\cdot\phi + \frac{m}{2}\,\phi\right)^2,
\end{aligned}
\tag{31}
$$

$$
\begin{aligned}
\mathcal{L} \longrightarrow \mathcal{L}_{(2)} = & -\frac{1}{2}\left[\frac{1}{2}\varepsilon_{\mu\nu\lambda\xi}\partial^\nu B^{\lambda\xi} \pm m\,\tilde{\phi}_\mu \pm \frac{1}{2}\partial_\mu\tilde{\phi}\right]^2 - \frac{m^2}{4}\,B_{\mu\nu}\,B^{\mu\nu} \\
& \pm \frac{m}{2}\left[\Phi^{\mu\nu} + \frac{1}{2}\varepsilon^{\mu\nu\rho\sigma}\tilde{\Phi}_{\rho\sigma}\right]B_{\mu\nu} - \frac{1}{4}\,\Phi_{\mu\nu}\,\Phi^{\mu\nu} + \frac{1}{4}\tilde{\Phi}_{\mu\nu}\,\tilde{\Phi}^{\mu\nu} \\
& + \frac{1}{2}\left[\partial^\nu B_{\nu\mu} \pm m\,\phi_\mu \pm \frac{1}{2}\partial_\mu\phi\right]^2 + \frac{1}{2}\left(\partial\cdot\tilde{\phi} - \frac{m}{2}\,\tilde{\phi}\right)^2 \\
& - \frac{1}{2}\left(\partial\cdot\phi - \frac{m}{2}\,\phi\right)^2,
\end{aligned}
\tag{32}
$$

where the mass dimensions of the fields have been taken into account, and we have taken into consideration *both* the signs that are present in (30) and chosen the constant numerical factor to be $1/2$. We shall corroborate the logic behind the choice of the terms containing $\phi$ and $\tilde{\phi}$ in the modified Lagrangian densities (31) and (32). We shall *also* dwell a bit on our choice of the factor $(1/2)$ in the kinetic and gauge-fixing terms that contain fields $\tilde{\phi}$ and $\phi$, respectively. The *latter* have been incorporated into $\mathcal{L}_{(1)}$ and $\mathcal{L}_{(2)}$ at appropriate places (e.g., the kinetic and gauge-fixing terms) with *proper* mass dimensions. It is worthwhile to mention that the signs of the last *two* terms, corresponding to the gauge-fixing of the axial-vector and polar-vector fields $\tilde{\phi}_\mu$ and $\phi_\mu$, respectively, are *fixed*, which leads to the EL-EoMs[5]: $(\Box + m^2)\,\phi_\mu = 0$, $(\Box + m^2)\,\tilde{\phi}_\mu = 0$, $(\Box + m^2)\,\phi = 0$, $(\Box + m^2)\,\tilde{\phi} = 0$.

At this juncture, we would like to point out that the generalization of the discrete duality symmetry transformations (10), namely:

$$
\begin{aligned}
& B_{\mu\nu} \longrightarrow \mp i\,\tilde{B}_{\mu\nu} \equiv \mp \frac{i}{2}\,\varepsilon_{\mu\nu\lambda\xi}\,B^{\lambda\xi}, \qquad \phi_\mu \longrightarrow \pm i\,\tilde{\phi}_\mu, \qquad \tilde{\phi}_\mu \longrightarrow \mp i\,\phi_\mu, \\
& \phi \longrightarrow \pm i\,\tilde{\phi}, \qquad \tilde{\phi} \longrightarrow \mp i\,\phi, \qquad\qquad B_{\mu\nu}\,B^{\mu\nu} \longrightarrow B_{\mu\nu}\,B^{\mu\nu},
\end{aligned}
\tag{33}
$$

is respected by the completely gauge-fixed Lagrangian densities $\mathcal{L}_{(1)}$ and $\mathcal{L}_{(2)}$, and *all* the fields (i.e., $B_{\mu\nu}$, $\phi_\mu$, $\tilde{\phi}_\mu$, $\phi$, $\tilde{\phi}$) satisfy the following Klein–Gordon equation[6]

$$
\begin{aligned}
& (\Box + m^2)\,B_{\mu\nu} = 0, \qquad (\Box + m^2)\,\phi_\mu = 0, \qquad (\Box + m^2)\,\tilde{\phi}_\mu = 0, \\
& (\Box + m^2)\,\phi = 0, \qquad (\Box + m^2)\,\tilde{\phi} = 0,
\end{aligned}
\tag{34}
$$

which is the signature of the completely (and correctly) gauge-fixed Lagrangian density. It should be noted that the mass term for the $B_{\mu\nu}$ field (i.e., $-(m^2/4)B_{\mu\nu}\,B^{\mu\nu}$) remains invariant under the transformation $[B_{\mu\nu} \longrightarrow \mp(i/2)\,\varepsilon_{\mu\nu\lambda\xi}\,B^{\lambda\xi}]$. The *latter* has its origin in the *self-duality* condition (cf. Equation (11)). This observation is *crucial* because it forces the whole theory to have a *single* mass parameter $m$. We point out that both the signs, chosen in the kinetic and gauge-fixing terms as well as in the *third* term of (31) and (32), are *allowed*, and they do *not* violate the Klein–Gordon equations in (34). It is very interesting to

highlight the following infinitesimal and continuous gauge transformations ($\delta_g$) for the basic fields of the Lagrangian density $\mathcal{L}_{(1)}$, namely

$$
\begin{aligned}
\delta_g B_{\mu\nu} &= -(\partial_\mu \Lambda_\nu - \partial_\nu \Lambda_\mu), & \delta_g \phi_\mu &= \pm(\partial_\mu \Lambda - m \Lambda_\mu), \\
\delta_g \Phi_{\mu\nu} &= \mp m(\partial_\mu \Lambda_\nu - \partial_\nu \Lambda_\mu), & \delta_g \phi &= \pm 2[(\partial \cdot \Lambda) + m \Lambda], \\
\delta_g [H_{\mu\nu\lambda}, \tilde{\phi}_\mu, \tilde{\phi}, \Phi_{\mu\nu}] &= 0,
\end{aligned}
\tag{35}
$$

which are nothing but the generalization of the gauge symmetry transformations (5). Under these transformations, we observe that the Lagrangian density $\mathcal{L}_{(1)}$ transforms as follows:

$$
\begin{aligned}
\delta_g \mathcal{L}_{(1)} = {} & \partial_\mu \left[ \mp m\, \varepsilon^{\mu\nu\lambda\xi} \Lambda_\nu \partial_\lambda \tilde{\phi}_\xi \right] \mp \left( \partial \cdot \phi + \frac{m}{2}\, \phi \right) [\Box + m^2] \Lambda \\
& - \left[ \partial_\nu B^{\nu\mu} \pm m\, \phi^\mu \mp \frac{1}{2} \partial^\mu \phi \right] [\Box + m^2] \Lambda_\mu.
\end{aligned}
\tag{36}
$$

Thus, it is crystal clear that if we impose the restrictions on the gauge transformation parameters as: $(\Box + m^2)\, \Lambda = 0$, $(\Box + m^2)\, \Lambda_\mu = 0$ from *outside*, the transformations (35) will become the *symmetry* transformations for the Lagrangian density $\mathcal{L}_{(1)}$. We shall see that, within the framework of the BRST approach to *this* theory, there will be *no* imposition of any kind of restriction from *outside* on the theory. We christen the infinitesimal and continuous transformations (35) as the *gauge* transformations because we observe that the *total* kinetic terms (i.e., $\delta_g H_{\mu\nu\lambda} = 0$, $\delta_g \tilde{\phi}_\mu = 0$, $\delta_g \tilde{\phi} = 0$, $\delta_g \tilde{\Phi}_{\mu\nu} = 0$), owing their origin *basically* to the exterior derivative $d = d\, x^\mu\, \partial_\mu$ (with $d^2 = 0$) of differential geometry [2–5], remain invariant. In addition to the gauge transformations (35), we have another set of infinitesimal and continuous transformations ($\delta_{dg}$) in the theory, namely;

$$
\begin{aligned}
\delta_{dg} B_{\mu\nu} &= -\varepsilon_{\mu\nu\lambda\xi} \partial^\lambda \Sigma^\xi, & \delta_{dg} \tilde{\phi}_\mu &= \pm(\partial_\mu \Omega - m \Sigma_\mu), \\
\delta_{dg} \tilde{\phi} &= \pm 2[\partial \cdot \Sigma + m \Omega], & \delta_{dg} \tilde{\Phi}_{\mu\nu} &= \mp m(\partial_\mu \Sigma_\nu - \partial_\nu \Sigma_\mu), \\
\delta_{dg} [\partial^\nu B_{\nu\mu}, \phi_\mu, \phi, \Phi_{\mu\nu}] &= 0,
\end{aligned}
\tag{37}
$$

which imply that the total gauge-fixing term (i.e., $\frac{1}{2}(\partial_\nu B^{\nu\mu} \pm m\, \phi^\mu \mp \frac{1}{2} \partial^\mu \phi)$), owing its origin primarily to the co-exterior derivative: $\delta = \pm * d *$, remains invariant. Here, the infinitesimal transformation parameters $\Sigma_\mu$ and $\Omega$ are the Lorentz axial vector and pseudo-scalar, respectively. We observe that the Lagrangian density $\mathcal{L}_{(1)}$ transforms under the infinitesimal and continuous transformations (37) as follows:

$$
\begin{aligned}
\delta_{dg} \mathcal{L}_{(1)} = {} & \partial_\mu \left[ \mp m\, \varepsilon^{\mu\nu\lambda\xi} \Sigma_\nu \partial_\lambda \phi_\xi \right] \pm \left( \partial \cdot \tilde{\phi} + \frac{m}{2}\, \tilde{\phi} \right) [\Box + m^2] \Omega \\
& + \left[ \frac{1}{2} \varepsilon^{\mu\nu\lambda\xi} \partial_\nu B_{\lambda\xi} \pm m\, \tilde{\phi}^\mu \mp \frac{1}{2} \partial^\mu \tilde{\phi} \right] [\Box + m^2] \Sigma_\mu,
\end{aligned}
\tag{38}
$$

which shows that, if we impose the conditions: $(\Box + m^2)\, \Sigma_\mu = 0$ and $(\Box + m^2)\, \Omega = 0$ from *outside*, the infinitesimal and continuous transformations (37) will become the *symmetry* transformations for the *completely* gauge-fixed Lagrangian density $\mathcal{L}_{(1)}$. We christen the infinitesimal transformations in (37) as the dual-gauge transformations ($\delta_{dg}$) because the gauge-fixing term for the $B_{\mu\nu}$ (and associated fields $\phi_\mu$ and $\phi$) remain invariant.

Before we end this section, we very *concisely* highlight a few key points connected with the continuous symmetries of the Lagrangian density $\mathcal{L}_{(2)}$ (cf. Equation (32)). In this context, it is very illuminating to point out that the following local, infinitesimal and continuous (dual-)gauge symmetry transformations [$\delta_{(d)g}$], namely;

$$
\begin{aligned}
\delta_{dg} B_{\mu\nu} &= -\varepsilon_{\mu\nu\lambda\xi} \partial^\lambda \Sigma^\xi, & \delta_{dg} \tilde{\phi}_\mu &= \pm(\partial_\mu \Omega - m \Sigma_\mu), \\
\delta_{dg} \tilde{\phi} &= \mp 2[\partial \cdot \Sigma + m \Omega], & \delta_{dg} \tilde{\Phi}_{\mu\nu} &= \mp m(\partial_\mu \Sigma_\nu - \partial_\nu \Sigma_\mu), \\
\delta_{dg} [\partial^\nu B_{\nu\mu}, \phi_\mu, \phi, \Phi_{\mu\nu}] &= 0,
\end{aligned}
\tag{39}
$$

$$\delta_g B_{\mu\nu} = -(\partial_\mu \Lambda_\nu - \partial_\nu \Lambda_\mu), \qquad \delta_g \phi_\mu = \pm (\partial_\mu \Lambda - m \Lambda_\mu),$$
$$\delta_g \Phi_{\mu\nu} = \mp m (\partial_\mu \Lambda_\nu - \partial_\nu \Lambda_\mu), \qquad \delta_g \phi = \mp 2 [(\partial \cdot \Lambda) + m \Lambda],$$
$$\delta_g [H_{\mu\nu\lambda}, \tilde{\phi}_\mu, \tilde{\phi}, \tilde{\Phi}_{\mu\nu}] = 0, \tag{40}$$

transform the Lagrangian density $\mathcal{L}_{(2)}$ as follows:

$$\delta_{dg} \mathcal{L}_{(2)} = \partial_\mu \left[ \mp m \, \varepsilon^{\mu\nu\lambda\xi} \Sigma_\nu \partial_\lambda \phi_\xi \right] \pm \left( \partial \cdot \tilde{\phi} - \frac{m}{2} \tilde{\phi} \right) [\Box + m^2] \, \Omega,$$
$$+ \left[ \frac{1}{2} \varepsilon^{\mu\nu\lambda\xi} \partial_\nu B_{\lambda\xi} \pm m \, \tilde{\phi}_\mu \pm \frac{1}{2} \partial_\mu \tilde{\phi} \right] [\Box + m^2] \, \Sigma_\mu$$

$$\delta_g \mathcal{L}_{(2)} = \partial_\mu \left[ \mp m \, \varepsilon^{\mu\nu\lambda\xi} \Lambda_\nu \partial_\lambda \tilde{\phi}_\xi \right] \mp \left( \partial \cdot \phi - \frac{m}{2} \phi \right) [\Box + m^2] \, \Lambda$$
$$- \left[ \partial_\nu B^{\nu\mu} \pm m \, \phi^\mu \pm \frac{1}{2} \partial^\mu \phi \right] [\Box + m^2] \, \Lambda_\mu. \tag{41}$$

It is evident that if we impose the restrictions

$$(\Box + m^2) \, \Sigma_\mu = 0, \qquad (\Box + m^2) \, \Lambda_\mu = 0,$$
$$(\Box + m^2) \, \Omega = 0, \qquad (\Box + m^2) \, \Lambda = 0, \tag{42}$$

on the dual-gauge transformation parameters $(\Sigma_\mu, \Omega)$ and the gauge transformation parameters $(\Lambda_\mu, \Lambda)$ from *outside*, we obtain the (dual-)gauge *symmetry* transformations (39) and (40) for the Lagrangian density $\mathcal{L}_{(2)}$. We note that the *outside* restrictions (36), (38), and (42) are *exactly* the same on the (dual-)gauge transformation parameters of our theory. Hence, when we elevate the Lagrangian densities $\mathcal{L}_{(1)}$ and $\mathcal{L}_{(2)}$ to their counterparts at the quantum *level* (within the framework of BRST formalism), we shall observe that the Faddeev–Popov ghost terms will be the *same* for the coupled (but equivalent) (anti-)BRST and (anti-)co-BRST invariant Lagrangian densities. The (anti-)ghost fields will *not* be restricted from *outside* for the *quantum* version of our theory within the ambit of BRST formalism (as the EoMs for the (anti-)ghost fields will take care of them).

## 5. Linearized Versions of the Lagrangian Densities: Auxiliary Fields and CF-Type Restrictions

We linearize the kinetic term for the $B_{\mu\nu}$ (and associated fields) and *all* the gauge-fixing terms by invoking the Nakanishi–Lautrup-type auxiliary fields. In this context, first of all, let us focus on the Lagrangian density $\mathcal{L}_{(1)}$, which can be written as:

$$\mathcal{L}_{(1)} \longrightarrow \mathcal{L}_{(b_1)} = \frac{1}{2} \mathcal{B}_\mu \mathcal{B}^\mu - \mathcal{B}^\mu \left[ \frac{1}{2} \varepsilon_{\mu\nu\lambda\xi} \partial^\nu B^{\lambda\xi} \pm m \, \tilde{\phi}_\mu \mp \frac{1}{2} \partial_\mu \tilde{\phi} \right]$$
$$- \frac{1}{4} \Phi_{\mu\nu} \Phi^{\mu\nu} + \frac{1}{4} \tilde{\Phi}_{\mu\nu} \tilde{\Phi}^{\mu\nu} \pm \frac{m}{2} B_{\mu\nu} \left[ \Phi^{\mu\nu} + \frac{1}{2} \varepsilon^{\mu\nu\rho\sigma} \tilde{\Phi}_{\rho\sigma} \right]$$
$$+ B^\mu \left[ \partial^\nu B_{\nu\mu} \pm m \, \phi_\mu \mp \frac{1}{2} \partial_\mu \phi \right] - \frac{1}{2} B^\mu B_\mu - \frac{m^2}{4} B_{\mu\nu} B^{\mu\nu}$$
$$+ B (\partial \cdot \phi + \frac{m}{2} \phi) + \frac{1}{2} B^2 - \mathcal{B} (\partial \cdot \tilde{\phi} + \frac{m}{2} \tilde{\phi}) - \frac{1}{2} \mathcal{B}^2. \tag{43}$$

The above Lagrangian density leads to the following equations of motion:

$$\mathcal{B}_\mu = \frac{1}{2} \varepsilon_{\mu\nu\lambda\xi} \partial^\nu B^{\lambda\xi} \pm m \, \tilde{\phi}_\mu \mp \frac{1}{2} \partial_\mu \tilde{\phi}, \qquad \mathcal{B} = -(\partial \cdot \tilde{\phi} + \frac{m}{2} \tilde{\phi}),$$
$$B_\mu = \partial^\nu B_{\nu\mu} \pm m \, \phi_\mu \mp \frac{1}{2} \partial_\mu \phi, \qquad B = -(\partial \cdot \phi + \frac{m}{2} \phi). \tag{44}$$

where the auxiliary fields $(\mathcal{B}_\mu, \, B_\mu, \, \mathcal{B}, \, B)$ are the Nakanishi–Lautrup auxiliary fields, which have been invoked for linearization purposes. For instance, the auxiliary field $\mathcal{B}_\mu$

has been invoked for the linearization of the *kinetic* term for the 2-form field $B_{\mu\nu}$ and associated fields. On the other hand, the auxiliary fields $(B_\mu, B, \mathcal{B})$ have been introduced to linearize the *gauge-fixing* terms for the $B_{\mu\nu}, \phi_\mu$ and $\tilde{\phi}_\mu$ fields, respectively. In an exactly similar fashion, we can linearize the Lagrangian density $\mathcal{L}_{(2)}$ by invoking a different set of Nakanishi–Lautrup-type auxiliary fields $(\bar{\mathcal{B}}_\mu, \bar{B}_\mu, \bar{\mathcal{B}}, \bar{B})$ as follows:

$$\mathcal{L}_{(2)} \longrightarrow \mathcal{L}_{(b_2)} = \frac{1}{2}\bar{\mathcal{B}}_\mu \bar{\mathcal{B}}^\mu + \bar{\mathcal{B}}^\mu \left[\frac{1}{2}\varepsilon_{\mu\nu\lambda\xi}\partial^\nu B^{\lambda\xi} \pm m\,\tilde{\phi}_\mu \pm \frac{1}{2}\partial_\mu\tilde{\phi}\right]$$
$$- \frac{1}{4}\Phi_{\mu\nu}\Phi^{\mu\nu} + \frac{1}{4}\tilde{\Phi}_{\mu\nu}\tilde{\Phi}^{\mu\nu} \pm \frac{m}{2}B_{\mu\nu}\left[\Phi^{\mu\nu} + \frac{1}{2}\varepsilon^{\mu\nu\rho\sigma}\tilde{\Phi}_{\rho\sigma}\right]$$
$$- \bar{B}^\mu\left[\partial^\nu B_{\nu\mu} \pm m\,\phi_\mu \pm \frac{1}{2}\partial_\mu\phi\right] - \frac{1}{2}\bar{B}^\mu\bar{B}_\mu - \bar{B}\left(\partial\cdot\phi - \frac{m}{2}\phi\right)$$
$$+ \frac{1}{2}\bar{B}^2 + \bar{\mathcal{B}}\left(\partial\cdot\tilde{\phi} - \frac{m}{2}\tilde{\phi}\right) - \frac{1}{2}\bar{\mathcal{B}}^2 - \frac{m^2}{4}B_{\mu\nu}B^{\mu\nu}. \tag{45}$$

The above Lagrangian density leads to the following equations of motion w.r.t. the Nakanishi–Lautrup-type auxiliary fields, namely;

$$\bar{\mathcal{B}}_\mu = -\left[\frac{1}{2}\varepsilon_{\mu\nu\lambda\xi}\partial^\nu B^{\lambda\xi} \pm m\,\tilde{\phi}_\mu \pm \frac{1}{2}\partial_\mu\tilde{\phi}\right], \quad \bar{\mathcal{B}} = \left(\partial\cdot\tilde{\phi} - \frac{m}{2}\tilde{\phi}\right),$$
$$\bar{B}_\mu = -\left[\partial^\nu B_{\nu\mu} \pm m\,\phi_\mu \pm \frac{1}{2}\partial_\mu\phi\right], \qquad \bar{B} = \left(\partial\cdot\phi - \frac{m}{2}\phi\right). \tag{46}$$

It is crystal clear that we can derive the following very useful and interesting relationships amongst the Nakanishi–Lautrup-type auxiliary fields *and* (pseudo-)scalar fields *from* the above equations of motion (44) and (46), namely;

$$\mathcal{B}_\mu + \bar{\mathcal{B}}_\mu \pm \partial_\mu\tilde{\phi} = 0, \qquad B + \bar{B} + m\,\phi = 0,$$
$$B_\mu + \bar{B}_\mu \pm \partial_\mu\phi = 0, \qquad \mathcal{B} + \bar{\mathcal{B}} + m\,\tilde{\phi} = 0, \tag{47}$$

which are nothing but the (anti-)BRST and (anti-)co-BRST invariant CF-type restrictions on our theory (see, e.g., [10,21] for details).

We end this section with the following remarks. First of all, the Lagrangian densities $\mathcal{L}_{(b_1)}$ and $\mathcal{L}_{(b_2)}$ have been derived in a completely different manner in our present endeavor if we compare our present method of derivation *against* the derivation in our earlier work [10], where we have exploited the method of *trial* and *error*. Second, the CF-type restrictions $B + \bar{B} + m\,\phi = 0$ and $\mathcal{B} + \bar{\mathcal{B}} + m\,\tilde{\phi} = 0$ are the *same* as in our earlier work [10,21], but the other *two* restrictions in (47) are different. Third, if we stick with the CF-type restrictions that have been derived from the superfield approach to BRST formalism in the context of 4D Abelian 2-form *massless* and *massive* gauge theories [18,21], we find that the other *two* restrictions of (47) are: $\mathcal{B}_\mu + \bar{\mathcal{B}}_\mu + \partial_\mu\tilde{\phi} = 0$ and $B_\mu + \bar{B}_\mu + \partial_\mu\phi = 0$. Hence, the $(\pm)$ signs associated with the (pseudo-)scalar fields (e.g., $\pm\frac{1}{2}\partial_\mu\phi$, $\pm\frac{1}{2}\partial_\mu\tilde{\phi}$) are *fixed*. As a consequence, we find that, in the Lagrangian density $\mathcal{L}_{(b_1)}$, we have only the *minus* signs for the scalar and pseudo-scalar fields (i.e., $-\frac{1}{2}\partial_\mu\phi$, $-\frac{1}{2}\partial_\mu\tilde{\phi}$) and the *plus* signs ($\frac{1}{2}\partial_\mu\phi$, $\frac{1}{2}\partial_\mu\tilde{\phi}$) for the exact expression for the Lagrangian density $\mathcal{L}_{(b_2)}$. Fourth, it is straightforward to note that the *duality* transformations (33) are now generalized in the following form:

$$B_{\mu\nu} \longrightarrow \mp\frac{i}{2}\varepsilon_{\mu\nu\lambda\xi}B^{\lambda\xi}, \quad \phi_\mu \longrightarrow \pm i\,\tilde{\phi}_\mu, \quad \tilde{\phi}_\mu \longrightarrow \mp i\,\phi_\mu \quad \phi \longrightarrow \pm i\,\tilde{\phi},$$
$$\tilde{\phi} \longrightarrow \mp i\,\phi, \quad \mathcal{B}_\mu \longrightarrow \mp i\,B_\mu, \quad B_\mu \longrightarrow \pm i\,\mathcal{B}_\mu, \quad B \longrightarrow \pm i\mathcal{B}, \quad \mathcal{B} \longrightarrow \mp i\,B,$$
$$\bar{\mathcal{B}}_\mu \longrightarrow \mp i\,\bar{B}_\mu, \quad \bar{B}_\mu \longrightarrow \pm i\,\bar{\mathcal{B}}_\mu, \quad \bar{B} \longrightarrow \pm i\bar{\mathcal{B}}, \quad \bar{\mathcal{B}} \longrightarrow \mp i\,\bar{B}. \tag{48}$$

Under the above discrete duality symmetry transformations, the coupled Lagrangian densities $\mathcal{L}_{(b_1)}$ and $\mathcal{L}_{(b_2)}$ are found to remain invariant *even* with the *fixed* choice of signs for the (pseudo-)scalar fields $\tilde{\phi}$ and $\phi$. Finally, in the next section, we shall take only the

simplest choices of the signs for the (pseudo-)scalar fields within the framework of BRST formalism where the Lagrangian density $\mathcal{L}_{(b_1)}$ will be generalized to incorporate into it the Faddeev–Popov ghost terms by following the standard technique [10,18,21].

## 6. Nilpotent (co-)BRST Invariant Lagrangian Density

We have generalized the Lagrangian densities $\mathcal{L}_{(b_1)}$ and $\mathcal{L}_{(b_2)}$ to their counterpart nilpotent (anti-)BRST and (anti-)co-BRST invariant Lagrangian densities $\mathcal{L}_{\mathcal{B}}$ and $\mathcal{L}_{\bar{\mathcal{B}}}$ that incorporate the Faddeev–Popov ghost terms. Such a set of coupled (but equivalent) Lagrangian densities have been written in our earlier works [10,21]. However, we shall focus on only *one* Lagrangian density and discuss the importance of discrete *duality* symmetry transformations (48) (and (54) below) which will connect the BRST transformations with the co-BRST transformations and vice versa. This kind of connection exists for the anti-BRST and anti-co-BRST symmetries, as well. However, we shall *not* dwell on the *latter* as it will be *only* an academic exercise. We would like to emphasize that, in our earlier works [10,21], such kinds of relationships have *not* been established where *only* the analogue of the Hodge duality operator (i.e., the set of discrete duality symmetry transformations) play a decisive role (along with the replacements: $s_b \Leftrightarrow s_d$). This observation is *totally* different from (60).

Towards the above goal in mind, we begin with the following (co-)BRST invariant Lagrangian density[7] (where $\mathcal{L}_{(b_1)} \longrightarrow \mathcal{L}_{\mathcal{B}}$) (see, e.g., [9,10,21])

$$
\begin{aligned}
\mathcal{L}_{\mathcal{B}} &= \frac{1}{2}\,\mathcal{B}_\mu\,\mathcal{B}^\mu - \mathcal{B}^\mu \left[ \frac{1}{2}\,\varepsilon_{\mu\nu\lambda\xi}\,\partial^\nu\,B^{\lambda\xi} + m\,\tilde{\phi}_\mu - \frac{1}{2}\,\partial_\mu\tilde{\phi} \right] - \frac{m^2}{4}\,B_{\mu\nu}\,B^{\mu\nu} \\
&\quad - \frac{1}{4}\,\Phi_{\mu\nu}\,\Phi^{\mu\nu} + \frac{1}{4}\,\tilde{\Phi}_{\mu\nu}\,\tilde{\Phi}^{\mu\nu} + \frac{m}{2}\,B_{\mu\nu}\left[ \Phi^{\mu\nu} + \frac{1}{2}\,\varepsilon^{\mu\nu\rho\sigma}\,\tilde{\Phi}_{\rho\sigma} \right] \\
&\quad + B^\mu \left[ \partial^\nu\,B_{\nu\mu} + m\,\phi_\mu - \frac{1}{2}\,\partial_\mu\,\phi \right] - \frac{1}{2}\,B^\mu\,B_\mu + B\left( \partial\cdot\phi + \frac{m}{2}\,\phi \right) \\
&\quad + \frac{1}{2}\,B^2 - \mathcal{B}\left( \partial\cdot\tilde{\phi} + \frac{m}{2}\,\tilde{\phi} \right) - \frac{1}{2}\,\mathcal{B}^2 - \frac{1}{2}\,\partial_\mu\,\bar{\beta}\,\partial^\mu\,\beta + \frac{m^2}{2}\,\bar{\beta}\,\beta \\
&\quad - (\partial_\mu\,\bar{C}_\nu - \partial_\nu\,\bar{C}_\mu)\,(\partial^\mu\,C^\nu) + (\partial_\mu\,\bar{C} - m\,\bar{C}_\mu)\,(\partial^\mu\,C - m\,C^\mu) \\
&\quad - \frac{1}{2}\left( \partial\cdot\bar{C} + m\,\bar{C} + \frac{\rho}{4} \right)\lambda - \frac{1}{2}\left( \partial\cdot C + m\,C - \frac{\lambda}{4} \right)\rho,
\end{aligned} \tag{49}
$$

where $(\bar{\beta})\,\beta$ are the *bosonic* (anti-)ghost fields with ghost numbers $(-2) + 2$, respectively, and $(\bar{C}_\mu)\,C_\mu$ are the *fermionic* $(\bar{C}_\mu\,C_\nu + C_\nu\,\bar{C}_\mu = 0,\ \bar{C}_\mu\,\bar{C}_\nu + \bar{C}_\nu\,\bar{C}_\mu = 0,\ C_\mu^2 = \bar{C}_\mu^2 = 0$, etc.) (anti-)ghost fields with ghost numbers $(-1) + 1$, respectively. In addition, we have Lorentz scalar *fermionic* $(C\,\bar{C} + \bar{C}\,C = 0,\ C^2 = \bar{C}^2 = 0$, etc.) (anti-)ghost fields with ghost numbers $(-1) + 1$, respectively. Our theory *also* contains the auxiliary *fermionic* $(\rho^2 = \lambda^2 = 0,\ \rho\,\lambda + \lambda\,\rho = 0)$ fields $(\rho)\,\lambda$ that carry the ghost numbers $(-1) + 1$, respectively.

The above Lagrangian density respects the following off-shell nilpotent $(s_b^2 = 0)$ BRST symmetry transformations $(s_b)$, namely;

$$
\begin{aligned}
&s_b\,B_{\mu\nu} = -(\partial_\mu\,C_\nu - \partial_\nu\,C_\mu), \quad s_b\,C_\mu = -\partial_\mu\,\beta, \quad s_b\bar{C}_\mu = B_\mu, \\
&s_b\,\bar{\beta} = -\rho, \quad s_b\,\phi_\mu = +(\partial_\mu C - m\,C_\mu), \quad s_b\,\bar{C} = B, \quad s_b\,\phi = +\lambda, \\
&s_b\,C = -m\,\beta, \quad s_b\,[H_{\mu\nu\lambda}, B, \lambda, \rho, B_\mu, \mathcal{B}_\mu, \beta, \mathcal{B}, \tilde{\phi}_\mu, \tilde{\phi}_{\mu\nu}, \tilde{\phi}] = 0,
\end{aligned} \tag{50}
$$

because the Lagrangian density $\mathcal{L}_{\mathcal{B}}$ transforms as [10]

$$
\begin{aligned}
s_b\,\mathcal{L}_{\mathcal{B}} &= \partial_\mu \Big[ -m\,\varepsilon^{\mu\nu\lambda\xi}\,\tilde{\phi}_\nu\,\partial_\lambda\,C_\xi - (\partial^\mu\,C^\nu - \partial^\nu\,C^\mu)\,B_\nu - \frac{1}{2}\,\lambda\,B^\mu \\
&\quad + B\,(\partial^\mu\,C - m\,C^\mu) + \frac{1}{2}\,\rho\,(\partial^\mu\,\beta) \Big],
\end{aligned} \tag{51}
$$

which implies that the action integral $S = \int d^4x\,\mathcal{L}_{\mathcal{B}}$ remains invariant (i.e., $s_b\,S = 0$) under the infinitesimal, continuous, and nilpotent BRST transformations (50). This happens

because of Gauss's divergence theorem, due to which *all* the physical fields vanish off as $x \longrightarrow \pm \infty$. In addition to $s_b$, the Lagrangian density $\mathcal{L}_\mathcal{B}$ *also* respects the infinitesimal, continuous, and nilpotent $[s_d^2 = 0]$ co-BRST (i.e., dual-BRST) transformations $(s_d)$ [10]:

$$
\begin{aligned}
s_d \, B_{\mu\nu} &= - \, \varepsilon_{\mu\nu\lambda\xi} \, \partial^\lambda \, \bar{C}^\xi, \quad s_d \, \bar{C}_\mu = - \, \partial_\mu \, \bar{\beta}, \quad s_d \, C_\mu = \mathcal{B}_\mu, \\
s_d \, \beta &= - \, \lambda, \quad s_d \, \tilde{\phi}_\mu = + (\partial_\mu \, \bar{C} - m \, \bar{C}_\mu), \quad s_d \, C = \mathcal{B}, \quad s_d \, \bar{C} = - \, m \, \bar{\beta}, \\
s_d \, \tilde{\phi} &= - \, \rho, \quad s_d \, [\partial^\nu \, B_{\nu\mu}, \, \mathcal{B}_\mu, \, B_\mu, \, \mathcal{B}, \, \phi_\mu, \, \Phi_{\mu\nu}, \, \phi, \, \bar{\beta}, \, \lambda, \, \rho] = 0.
\end{aligned} \tag{52}
$$

It is straightforward to check that $\mathcal{L}_\mathcal{B}$ transforms, under $(s_d)$, as the total spacetime derivative in the four $(3 + 1)$-dimensional (4D) spacetime, namely;

$$
\begin{aligned}
s_d \, \mathcal{L}_\mathcal{B} = \; &\partial_\mu \Big[ - m \, \varepsilon^{\mu\nu\lambda\xi} \, \phi_\nu \, \partial_\lambda \, \bar{C}_\xi + (\partial^\mu \, \bar{C}^\nu - \partial^\nu \, \bar{C}^\mu) \, \mathcal{B}_\nu - \frac{1}{2} \, \rho \, \mathcal{B}^\mu \\
&- \; (\partial^\mu \, \bar{C} - m \, \bar{C}^\mu) \, \mathcal{B} + \frac{1}{2} \, \lambda \, (\partial^\mu \, \bar{\beta}) \Big].
\end{aligned} \tag{53}
$$

As a consequence of the above observation, we find that the action integral $S = \int d^4 x \, \mathcal{L}_\mathcal{B}$ remains invariant (i.e., $s_d \, S = 0$) under the co-BRST symmetry transformation $s_d$ for *all* the physical fields that vanish off as $x \longrightarrow \pm \infty$.

In addition to discrete duality symmetry transformations (48) in the bosonic (i.e., non-ghost) *sector* of the Lagrangian density $\mathcal{L}_\mathcal{B}$, we have the following *discrete* symmetry transformations in the ghost-sector[8]:

$$
\begin{aligned}
C_\mu &\longrightarrow \pm i \, \bar{C}_\mu, \quad \bar{C}_\mu \longrightarrow \pm i \, C_\mu, \quad C \longrightarrow \pm i \, \bar{C}, \quad \bar{C} \longrightarrow \pm i \, C, \\
\rho &\longrightarrow \mp i \, \lambda, \quad \lambda \longrightarrow \mp i \, \rho, \quad \beta \longrightarrow \pm i \, \bar{\beta}, \quad \bar{\beta} \longrightarrow \mp i \, \beta.
\end{aligned} \tag{54}
$$

Under the *full* discrete duality symmetry transformations (48) and (54), it can be checked that the (co-)BRST symmetry transformations (32) and (50) are interconnected. To corroborate this claim, let us begin with $s_b \, B_{\mu\nu} = - (\partial_\mu \, C_\nu - \partial_\nu \, C_\mu)$. If we apply the discrete symmetry transformations (48) and (54) on *it* and take the replacement: $s_b \longrightarrow s_d$, we obtain the following explicit relationship:

$$
s_b \, (\ast \, B_{\mu\nu}) = - \ast (\partial_\mu \, C_\nu - \partial_\nu \, C_\mu) \quad \Longrightarrow \quad s_d \, B_{\mu\nu} = - \, \varepsilon_{\mu\nu\lambda\xi} \, \partial^\lambda \, \bar{C}^\xi, \tag{55}
$$

where $\ast$ is nothing but the *full* discrete duality symmetry transformations (48) *plus* (54). In other words, we have obtained the co-BRST symmetry transformation $s_d$ operating on $B_{\mu\nu}$ field *from* the operation of $s_b$ on $B_{\mu\nu}$. In an exactly similar fashion, we note the following (with the replacement: $s_d \longrightarrow s_b$), for the transformations $s_d \, B_{\mu\nu} = - \, \varepsilon_{\mu\nu\lambda\xi} \, \partial^\nu \, \bar{C}^\xi$, namely;

$$
s_d \, (\ast \, B_{\mu\nu}) = - \ast \varepsilon_{\mu\nu\lambda\xi} \, (\partial^\lambda \, \bar{C}^\xi) \quad \longrightarrow \quad s_b \, B_{\mu\nu} = - (\partial_\mu \, C_\nu - \partial_\nu \, C_\mu), \tag{56}
$$

where, once again, the $\ast$ operation is nothing but the *total* discrete duality symmetry transformations (48) *plus* (54). This observation is *not* limited only to the *bosonic* antisymmetric tensor gauge field. To corroborate this assertion, let us focus on the symmetry transformation: $s_b \, \phi_\mu = + (\partial_\mu \, C - m \, C_\mu)$ on a bosonic *vector* field $(\phi_\mu)$. By exploiting the strength of the *full* discrete *duality* symmetry transformations (48) *plus* (54), we observe the following transformations on the axial-vector field (with input: $s_b \longrightarrow s_d$), namely;

$$
s_b \, (\ast \, \phi_\mu) = + \ast (\partial_\mu \, C - m \, C_\mu) \quad \Longrightarrow \quad s_d \, \tilde{\phi}_\mu = + (\partial_\mu \, \bar{C} - m \, \bar{C}_\mu). \tag{57}
$$

This happens because, under discrete duality symmetry transformations (48), we have: $\phi_\mu \to \pm i \, \tilde{\phi}_\mu$ and $\tilde{\phi}_\mu \to \mp i \, \phi_\mu$. In an exactly,similar fashion, we obtain the reciprocal symmetry transformations as follows (with inputs: $s_d \longrightarrow s_b$ and use of the discrete duality symmetry transformations), namely;

$$
s_d \, (\ast \, \tilde{\phi}_\mu) = + \ast (\partial_\mu \, \bar{C} - m \, \bar{C}_\mu) \quad \Longrightarrow \quad s_b \, \phi_\mu = + (\partial_\mu \, C - m \, C_\mu). \tag{58}
$$

The above kind of exercise can be repeated with *all* the fields of our theory. We observe that the discrete duality symmetry transformations (48) and (54) are the generalization of our basic discrete duality *symmetry* transformations $(B_{\mu\nu} \longrightarrow \mp (i/2) \, \varepsilon_{\mu\nu\lambda\xi} \, B^{\lambda\xi}, \; \phi_\mu \longrightarrow \pm i \, \tilde{\phi}_\mu, \; \tilde{\phi}_\mu \longrightarrow \mp i \, \phi_\mu)$ of the *modified* Stückelberg formalism (cf. Equations (9) and (10)). To complete our present discussion, let us focus on a transformation on a *fermionic* field $s_d \, \bar{C} = - \, m \, \bar{\beta}$. Using the strength of the discrete *duality* symmetry transformations (54), we obtain the following (with the input: $s_d \longrightarrow s_b$), namely;

$$s_d \, [* \, (\bar{C})] = - \, m * \bar{\beta} \quad \Longrightarrow \quad s_b \, C = - \, m \, \beta. \tag{59}$$

Thus, we are able to obtain the BRST symmetry transformation: $s_b \, C = - \, m \, \beta$ from the co-BRST symmetry transformation: $s_d \, \bar{C} = - \, m \, \bar{\beta}$ by exploiting the strength of the discrete *duality* symmetry transformations (54). Hence, our observation is true for *fermionic* field, as well. It goes without saying that, repeating the same procedure, we can obtain: $s_d \, \bar{C} = - \, m \, \bar{\beta}$ from the given BRST symmetry transformation: $s_b \, C = - \, m \, \beta$. Thus, the discrete *duality* symmetry transformations (48) *plus* (54) connect the BRST and co-BRST transformations for the *bosonic* as well as the *fermionic* fields of our theory.

We end this section with the following remarks. First, the discrete duality symmetry transformations (48) and (54) are able to provide a connection between the symmetry transformations $s_b$ and $s_d$. Second, it can be seen that the interplay of the discrete and continuous symmetry transformations provides [10] the physical realization of $\delta = \pm * d *$ that exist [2–5] between the (co-)exterior derivatives $((\delta)d)$ of differential geometry. This interesting and beautiful relationship between $s_d$ and $s_b$ is[9]

$$s_d = \pm * s_b *, \tag{60}$$

where $*$ is nothing but the complete set of discrete *duality* symmetry transformations (48) and (54). Third, despite the above connections between the BRST and co-BRST symmetry transformations in the language of the symmetry properties of our theory, these symmetries are *independent* of each-other in the *same* manner as *do* the exterior $(d)$ and co-exterior $(\delta)$ derivatives of differential geometry [2–5] even though these derivatives are connected with each other by the relationship: $\delta = \pm * d *$. Finally, it can be seen that the *exactly* similar kinds of relationships exist between the nilpotent anti-co-BRST symmetry and anti-BRST symmetry transformations that exist for the Lagrangian density $\mathcal{L}_{\bar{B}}$ (which turns out to be the generalization of the Lagrangian density $\mathcal{L}_{b_2}$ (see e.g., [9,10] for details)).

## 7. Conclusions

The Stückelberg-modified massive 4D free Abelian 2-form theory has already been proven to be a *massive* model of Hodge theory [10], where its discrete and continuous symmetry transformations (and corresponding conserved charges) have been shown to provide the physical realizations of the de Rham cohomological operators [2–5] of the differential geometry at the *algebraic* level within the framework of BRST formalism [10]. However, the *full* coupled (but equivalent) Lagrangian densities of this theory have been obtained by the *trial* and *error* method. In our present investigation, we have theoretically derived the *correct* forms of the coupled (but equivalent) Lagrangian densities. To be precise, we have concentrated *only* on the (co-)BRST invariant Lagrangian density (cf. Section 6) for the sake of brevity but indicated the theoretical methodology for the derivation of the coupled (but equivalent) Lagrangian densities that respect *six* continuous and a couple of *useful* discrete duality symmetry transformations (see, e.g., [10]) within the framework of BRST formalism. The above set of symmetries entail upon this model (i.e., the 4D massive Abelian 2-form theory) to become a massive field-theoretic example of Hodge theory.

One of the key results of our present investigation is the modification (cf. Equations (7) and (9)) of the Stückelberg formalism on the 4D flat Minkowskian spacetime manifold where the ideas from the differential geometry have played very important roles. It has been demonstrated that the *modified* SF remains form-invariant under the discrete duality

symmetry transformations (cf. Equation (10)), whose generalizations (cf. Equations (48) and (54)), within the realm of BRST formalism, provide the physical realizations of the Hodge duality $*$ operator of the differential geometry. As the gauge-fixed Lagrangian density (29) remains invariant under the discrete duality symmetry transformations (10), in an exactly similar fashion, the (co-)BRST invariant Lagrangian density (49) remains invariant under the generalization of the discrete *duality* symmetry transformations (cf. Equation (10)): (*i*) to Equation (48) in the non-ghost sector and (*ii*) to Equation (54) in the ghost-sector of the Lagrangian density (49). In addition, we have been able to establish a *direct* connection between the BRST and co-BRST symmetry transformations (i.e., $s_b \leftrightarrow s_d$) due to the existence of the discrete duality symmetry transformations (48) and (54), which is a *novel* result in our present investigation. The *latter* symmetry transformations ($s_b$ and $s_d$) also play an important role [10] in providing the analogue of relationship: $\delta = \pm * d *$ in the terminology of nilpotent symmetry transformations of our present *massive* 4D theory (cf. Equation (60)).

It is worthwhile to mention that the modified SF (cf. Equation (9)) is invariant under the discrete duality symmetry transformations (10), and they lead to the combination of the polar vector and axial-vector fields ($\phi_\mu$ and $\tilde{\phi}_\mu$) in the form: $\partial_\mu \phi_\nu - \partial_\nu \phi_\mu + \varepsilon_{\mu\nu\lambda\xi} \partial^\lambda \tilde{\phi}^\xi$. Exactly the *same* combination has been taken by Zwanziger [22] in the description of the (electromagnetic global duality invariant) 4D Maxwell theory of electrodynamics with *double* potentials with the field strength tensor as: $F_{\mu\nu} = \partial_\mu V_\nu - \partial_\nu V_\mu + \varepsilon_{\mu\nu\lambda\xi} \partial^\lambda A^\xi$, where $V_\mu$ and $A_\mu$ are the polar vector and axial-vector potentials, respectively. We have discussed the *local* duality invariance [23] of the Maxwell theory with *these* potentials and shown the existence of an axial photon which mediates the spin-spin *universal* long-range interaction (see, e.g., [23,24] for details). However, we have *not* discussed the applications of the axial-vector potential $A_\mu$ in the context of dark energy/dark matter. On the contrary, a close and careful look at the Lagrangian densities (31) and (32) demonstrates that the fields $\tilde{\phi}_\mu$ and $\tilde{\phi}$ turn up with *negative* kinetic terms in our theory, which are *interesting* in the sense that they belong to a class of *exotic* fields that are supposed to be one of the possible set of candidates for the dark matter/dark energy [19,20] *and* the "phantom" and/or "ghost" fields in the context of the modern developments in the cyclic, bouncing and self-accelerated cosmological models of the Universe [13–15], which take care of the modern experimental observation of the accelerated expansion of the Universe.

In a set of very nice works [25–27], the Stückelberg-modified (SUSY) quantum electrodynamics and other aspects of the (non-)interacting Abelian gauge theories have been considered, where an *ultralight* dark matter candidate has been proposed (and the Stückelberg boson has been able to cure the infrared problem in QED). It will be an interesting idea to apply our BRST approach to the examples that have been considered in [25–27]. Furthermore, we have already established that the 6D Abelian 3-form gauge theory is a model of Hodge theory within the ambit of BRST formalism [9]. It will be a nice future endeavor to extend our understandings of the 2D Stückelberg-modified Proca (i.e., the *massive* Abelian 1-form) theory [8] as well as our *present* work (on the Stückelberg-modified *massive* 4D free Abelian 2-form theory) to study the Stückelberg-modified *massive* 6D Abelian 3-form theory within the framework of BRST formalism. In a very recent work [28], a prototype system of first-class constraints and various kinds of BRST-type symmetries and their relationships have been established. It will be interesting to see weather the brand-*new* BRST-type symmetries (that have been pointed out in [28]) can be accommodated within the framework of field-theoretic models of Hodge theory. We are involved with these ideas at present, and we shall report on our progress *elsewhere* in our future publication(s).

**Author Contributions:** R.P.M. led the analysis, conceptualization and preparation of the manuscript. A.K.R. contributed to the analysis, printing and preparation of the manuscript. All authors have read and agreed to the published version of the manuscript.

**Funding:** One of us (AKR) thankfully acknowledges the financial support from the *BHU fellowship program* of the Banaras Hindu University (BHU), Varanasi, under which the present research work has been carried out.

**Data Availability Statement:** Not available.

**Acknowledgments:** Fruitful discussions with S. Kumar, B. Chauhan and A. Tripathi are gratefully acknowledged. The authors dedicate their present work, very humbly and respectfully, to the memory of T. Pradhan, who was the Ph.D. advisor to one (RPM) of them and who passed away in the recent past. Fruitful comments by our esteemed reviewers are also thankfully acknowledged.

**Conflicts of Interest:** The authors declare no conflict of interest.

## Appendix A. On Modified 2D Proca Theory

For our present paper to be self-contained, we dwell a bit on the free *massive* 2D Abelian 1-form (i.e., Proca) theory, which has been at the *heart* of our present investigation on the free *massive* 4D Abelian 2-form theory. We start off with the Proca Lagrangian density $[\mathcal{L}_{(P)}]$ for a vector boson ($A_\mu$) with rest mass $m$ as follows (see, e.g., [17])

$$\mathcal{L}_{(P)} = -\frac{1}{4} F_{\mu\nu} F^{\mu\nu} + \frac{m^2}{2} A_\mu A^\mu, \tag{A1}$$

where the 2-form $F^{(2)} = d A^{(1)} = [(d x^\mu \wedge d x^\nu)/2!] F_{\mu\nu}$ defines the field strength tensor: $F_{\mu\nu} = \partial_\mu A_\nu - \partial_\nu A_\mu$ for the vector field $A_\mu$ that is defined through an Abelian 1-form ($A^{(1)} = d x^\mu A_\mu$). Here the symbol $d = d x^\mu \partial_\mu$ (with $d^2 = 0$) stands for the exterior derivative of differential geometry [2–5]. The standard Stückelberg formalism (valid in any arbitrary D-dimensional spacetime) is modified in the 2D case as (see, e.g., [8] for details)

$$A_\mu \longrightarrow A_\mu \mp \frac{1}{m} (\partial_\mu \phi + \varepsilon_{\mu\nu} \partial^\nu \tilde{\phi}), \tag{A2}$$

where $\phi$ is a pure-scalar field, and $\tilde{\phi}$ is a pseudo-scalar field in 2D spacetime, which is endowed with the Levi–Civita tensor $\varepsilon_{\mu\nu}$ (with $\varepsilon_{01} = \varepsilon^{10} = +1$, $\varepsilon_{\mu\nu} \varepsilon^{\mu\nu} = -2!$, $\varepsilon_{\mu\nu} \varepsilon^{\mu\rho} = -1! \delta^\rho_\nu$, $E = -\varepsilon^{\mu\nu} \partial_\mu A_\nu = F_{01}$, etc.). It can be readily checked that the *modified* 2D Stückelberg formalism is *invariant* under the discrete symmetry transformations: $A_\mu \to \mp i \varepsilon_{\mu\nu} A^\nu$, $\phi \to \mp i \tilde{\phi}$, $\tilde{\phi} \to \mp i \phi$ which play a very important role in establishing a relationship with the Hodge duality $*$ operator of the differential geometry (see, e.g., [8]).

We observe that, under the *modified* Stückelberg formalism (A2), the field-strength tensor transforms as (see, e.g., [8])

$$F_{\mu\nu} \longrightarrow F_{\mu\nu} \mp \frac{1}{m} (\varepsilon_{\nu\rho} \partial_\mu - \varepsilon_{\mu\rho} \partial_\nu) (\partial^\rho \tilde{\phi}). \tag{A3}$$

We can introduce a notation $\Sigma_{\mu\nu} = (\varepsilon_{\mu\rho} \partial_\nu - \varepsilon_{\nu\rho} \partial_\mu) (\partial^\rho \tilde{\phi})$ to re-express the above transformation for the field strength tensor as follows

$$F_{\mu\nu} \longrightarrow F_{\mu\nu} \pm \frac{1}{m} \Sigma_{\mu\nu}, \tag{A4}$$

which leads to the following transformation for the kinetic term ($-(1/4) F_{\mu\nu} F^{\mu\nu}$) of the Proca (i.e., massive Abelian 1-form) theory, namely;

$$-\frac{1}{4} F^{\mu\nu} F_{\mu\nu} \longrightarrow -\frac{1}{4} F^{\mu\nu} F_{\mu\nu} \mp \frac{1}{2m} F^{\mu\nu} \Sigma_{\mu\nu} - \frac{1}{4m^2} \Sigma^{\mu\nu} \Sigma_{\mu\nu}. \tag{A5}$$

It is straightforward to note that the *second* and *third* terms, on the r.h.s. of (A5), are *higher* order derivative terms for a 2D theory of a vector boson. This is due to the fact that there are *three* and *four* derivatives in the *second* and *third* terms, respectively. This is clear from the transformation for the field strength tensor (cf. Equation (A5)) under (A2) .

We can remove the higher derivative terms by exploiting the on-shell conditions: $\partial_\mu F^{\mu\nu} + m^2 A^\nu = 0$ and $(\Box + m^2)\tilde{\phi} = 0$. The *latter* implies that the on-shell condition $(\Box + m^2)\partial^\mu \tilde{\phi} = 0$ is *also* true. The second term, on the r.h.s. of (A5), can be explicitly expressed as follows:

$$\mp \frac{1}{2\,m}\, F^{\mu\nu} \left(\varepsilon_{\mu\rho}\, \partial_\nu - \varepsilon_{\nu\rho}\, \partial_\mu\right) (\partial^\rho\, \tilde{\phi}). \tag{A6}$$

Dropping the total spacetime derivative terms, we note that we have the following explicit form of (A6), namely;

$$\pm \frac{1}{2\,m}\, (\partial_\nu\, F^{\mu\nu})\, \varepsilon_{\mu\rho}\, (\partial^\rho\, \tilde{\phi}) \mp \frac{1}{2\,m}\, (\partial_\mu\, F^{\mu\nu})\, \varepsilon_{\nu\rho}\, (\partial^\rho\, \tilde{\phi}), \tag{A7}$$

where *both* the terms are equal, and they lead to the following (due to the use of $E = -\varepsilon^{\mu\nu} \partial_\mu A_\nu$ and on-shell condition: $\partial_\mu F^{\mu\nu} = -m^2 A^\nu$), namely;

$$\pm\, m\, A^\nu\, \varepsilon_{\nu\rho}\, \partial^\rho\, \tilde{\phi} \;\equiv\; \pm\varepsilon^{\rho\nu}\, (\partial_\rho\, A_\nu)\, \tilde{\phi} \;\equiv\; \mp m\, E\, \tilde{\phi}. \tag{A8}$$

In the above, we have dropped a total spacetime derivative term (as it is a part of the Lagrangian density , and its presence *does* not change the dynamics of our 2D Stückelberg-modified Proca theory with $\phi$ and $\tilde{\phi}$ as the compensating fields).

We concentrate now on the *third* term (with four derivatives) on the r.h.s. of Equation (A5), which is explicitly expressed as:

$$-\frac{1}{4\,m^2}\, \Sigma^{\mu\nu}\, \Sigma_{\mu\nu} \;=\; -\frac{1}{4\,m^2}\, \left[(\varepsilon^{\mu\rho}\, \partial^\nu - \varepsilon^{\nu\rho}\, \partial^\mu)\, (\partial_\rho\, \tilde{\phi})\right]\left[(\varepsilon_{\mu\sigma}\, \partial_\nu - \varepsilon_{\nu\sigma}\, \partial_\mu)\, (\partial^\sigma\, \tilde{\phi}). \tag{A9}$$

The above expression, belonging to the Lagrangian density, leads to the following expression (modulo the total spacetime derivatives), namely;

$$-\frac{2}{4\,m^2}\, \left[\varepsilon^{\mu\rho}\, \partial^\nu\, (\partial_\rho\, \tilde{\phi})\, \varepsilon_{\mu\sigma}\, \partial_\nu\, (\partial^\sigma\, \tilde{\phi})\right] \;\equiv\; \frac{1}{2\,m^2}\, \partial^\nu\, (\partial_\sigma\, \tilde{\phi})\partial_\nu\, (\partial^\sigma\, \tilde{\phi}), \tag{A10}$$

where we have used $\varepsilon^{\mu\rho}\, \varepsilon_{\mu\sigma} = -\delta^\rho_\sigma$. Dropping, once again, the total spacetime derivative term, we obtain the following:

$$\frac{1}{2}\, \partial_\mu\, \tilde{\phi}\, \partial^\mu\, \tilde{\phi} \;\equiv\; +\frac{1}{2}\, m^2\, \tilde{\phi}^2, \tag{A11}$$

where we have used the on-shell conditions: $(\Box + m^2)\, \partial^\sigma\, \tilde{\phi} = 0$, $(\Box + m^2)\, \tilde{\phi} = 0$. Since the field-strength tensor $F_{\mu\nu}$ has only *one* non-vanishing component in 2D (which is nothing but the pseudo-scalar electric field $E = F_{01} = -\varepsilon^{\mu\nu} \partial_\mu A_\nu$), we note that the explicit form of (A5), with the help of (A8) and (A11), is as follows:

$$-\frac{1}{4}\, F_{\mu\nu}\, F^{\mu\nu} \;\longrightarrow\; \frac{1}{2}\, E^2 \mp m\, E\, \tilde{\phi} + \frac{1}{2}\, m^2\, \tilde{\phi}^2 \equiv \frac{1}{2}\, (E \mp m\, \tilde{\phi})^2, \tag{A12}$$

which has been derived in a different manner in our earlier work [8]. It is straightforward to note that the *mass* term of (A1) transforms, under the redefinition (A2), as follows

$$\begin{aligned}\frac{m^2}{2}\, A_\mu\, A^\mu \;\longrightarrow\; &\frac{m^2}{2}\, A_\mu\, A^\mu \mp m\, A_\mu\, \partial^\mu\, \phi + \frac{1}{2}\, \partial_\mu\, \phi\, \partial^\mu\, \phi \\ &-\frac{1}{2}\, \partial_\mu\, \tilde{\phi}\, \partial^\mu\, \tilde{\phi} \pm m\, E\, \tilde{\phi}, \end{aligned} \tag{A13}$$

modulo some *total* spacetime derivative terms. Here, we have used $E = -\varepsilon^{\mu\nu} \partial_\mu A_\nu$. Thus, the total Lagrangian density for the modified version of 2D Proca theory is

$$\begin{aligned}\mathcal{L}^{(2D)}_{(P)} \;=\; &\frac{1}{2}\, (E \mp m\, \tilde{\phi})^2 \pm m\, E\, \tilde{\phi} - \frac{1}{2}\, \partial_\mu\, \tilde{\phi}\, \partial^\mu\, \tilde{\phi} + \frac{m^2}{2}\, A_\mu\, A^\mu \\ &\mp m\, A_\mu\, \partial^\mu\, \phi + \frac{1}{2}\, \partial_\mu\, \phi\, \partial^\mu\, \phi, \end{aligned} \tag{A14}$$

which has been taken into account in our earlier works [8,11]. It is important to point out that the kinetic terms of the pure-scalar and pseudo-scalar fields have *positive* and *negative* signs, respectively. The *latter* (i.e., the pseudo-scalar) field is interesting from the point of view of the fact that it provides a possible candidate for the dark matter/dark energy. Such *exotic* fields are also useful in the context of cyclic, bouncing and self-accelerated cosmological models of the Universe [13–15], where *these* (i.e., fields with negative kinetic terms) have been called "phantom" and/or "ghost" fields.

We end this Appendix with the *final* comment that one can add the gauge-fixing term ($\mathcal{L}_{gf}$) to the above Lagrangian density (A14) in the 't Hooft gauge as follows [17]:

$$
\begin{aligned}
\mathcal{L}_{(S)}^{(2D)} + \mathcal{L}_{(gf)} \;=\;& \tfrac{1}{2}\left(E \mp m\,\tilde{\phi}\right)^2 \pm m\,E\,\tilde{\phi} - \tfrac{1}{2}\,\partial_\mu\tilde{\phi}\,\partial^\mu\tilde{\phi} + \tfrac{m^2}{2}A_\mu A^\mu \\
& \mp A_\mu\,\partial^\mu\phi + \tfrac{1}{2}\,\partial_\mu\phi\,\partial^\mu\phi - \tfrac{1}{2}\left(\partial \cdot A \pm m\,\phi\right)^2,
\end{aligned}
\tag{A15}
$$

which respects the discrete duality symmetry transformations on the *basic* fields of the theory as: $A_\mu \to \mp i\,\varepsilon_{\mu\nu}A^\nu$, $\phi \to \mp i\,\tilde{\phi}$, $\tilde{\phi} \to \mp i\,\phi$. The Lagrangian density (A15) has been taken into account for the BRST analysis in our earlier works [8,11], where we have proven that the *modified* 2D Proca theory is a field-theoretic example for the Hodge theory.

## Appendix B. On the Generalized Nilpotent (co-)BRST Symmetries and Uniqueness of the Lagrangian Density

The central purpose of our present Appendix is to generalize the *classical* (dual-)gauge symmetry transformations (37) and (35), respectively, to their counterpart *quantum* (co-)BRST symmetry transformations for the appropriate *generalized* form of the (co-)BRST invariant Lagrangian density (which is more general than the Lagrangian density (49)). First of all, we generalize the *classical* gauge symmetry transformations (35) to the following off-shell nilpotent ($s_b^2 = 0$) *quantum* BRST symmetry transformations, namely;

$$
\begin{aligned}
s_b\,B_{\mu\nu} &= -\left(\partial_\mu C_\nu - \partial_\nu C_\mu\right), & s_b\,C_\mu &= -\partial_\mu\beta, & s_b\,\bar{C}_\mu &= B_\mu, \\
s_b\,\phi_\mu &= \pm\left(\partial_\mu C - m\,C_\mu\right), & s_b\,C &= -m\,\beta, & s_b\,\bar{C} &= B, \\
s_b\,\Phi_{\mu\nu} &= \mp\,m\left(\partial_\mu C_\nu - \partial_\nu C_\mu\right) & s_b\,\phi &= \pm\lambda, & s_b\,\bar{\beta} &= \mp\,\rho, \\
& \multicolumn{5}{l}{s_b\left[H_{\mu\nu\lambda},\,\rho,\,\lambda,\,\beta,\,\mathcal{B}_\mu,\,B_\mu,\,B,\,\tilde{\phi},\,\tilde{\phi}_\mu,\,\tilde{\Phi}_{\mu\nu}\right] = 0,}
\end{aligned}
\tag{A16}
$$

which transform the following *generalized* (co-)BRST invariant Lagrangian density ($\mathcal{L}_{\mathcal{B}}^{(g)}$), with appropriate ($\pm$) signs, namely;

$$
\begin{aligned}
\mathcal{L}_{\mathcal{B}}^{(g)} \;=\;& \tfrac{1}{2}\,\mathcal{B}_\mu\,\mathcal{B}^\mu - \mathcal{B}^\mu\left[\tfrac{1}{2}\,\varepsilon_{\mu\nu\lambda\xi}\left(\partial^\nu B^{\lambda\xi}\right) + m\,\tilde{\phi}_\mu - \tfrac{1}{2}\,\partial_\mu\tilde{\phi}\right] - \tfrac{m^2}{4}\,B_{\mu\nu}\,B^{\mu\nu} \\
& -\tfrac{1}{4}\,\Phi_{\mu\nu}\,\Phi^{\mu\nu} + \tfrac{1}{4}\,\tilde{\Phi}_{\mu\nu}\,\tilde{\Phi}^{\mu\nu} \pm \tfrac{m}{2}\,B_{\mu\nu}\left[\Phi^{\mu\nu} + \tfrac{1}{2}\varepsilon^{\mu\nu\lambda\xi}\,\tilde{\Phi}_{\lambda\xi}\right] \\
& +B^\mu\left[\left(\partial^\nu B_{\nu\mu}\right) + m\,\phi_\mu - \tfrac{1}{2}\,\partial_\mu\phi\right] - \tfrac{1}{2}\,B^\mu\,B_\mu + B\left(\partial \cdot \phi + \tfrac{m}{2}\,\phi\right) \\
& +\tfrac{1}{2}\,B^2 - \mathcal{B}\left(\partial \cdot \tilde{\phi} + \tfrac{m}{2}\,\tilde{\phi}\right) - \tfrac{1}{2}\,\mathcal{B}^2 - \tfrac{1}{2}\,\partial_\mu\bar{\beta}\,\partial^\mu\beta + \tfrac{m^2}{2}\,\bar{\beta}\,\beta \\
& -\left(\partial_\mu\bar{C}_\nu - \partial_\nu\bar{C}_\mu\right)\left(\partial^\mu C^\nu\right) \pm \left(\partial_\mu\bar{C} - m\,\bar{C}_\mu\right)\left(\partial^\mu C - m\,C^\mu\right) \\
& \mp\tfrac{1}{2}\left(\partial \cdot \bar{C} + m\,\bar{C} + \tfrac{\rho}{4}\right)\lambda \mp \tfrac{1}{2}\left(\partial \cdot C + m\,C - \tfrac{\lambda}{4}\right)\rho,
\end{aligned}
\tag{A17}
$$

to the total spacetime derivative on the 4D Minkowskian spacetime manifold, as:

$$
\begin{aligned}
s_b\,\mathcal{L}_{\mathcal{B}}^{(g)} \;=\;& \partial_\mu\Big[\mp m\,\varepsilon^{\mu\nu\lambda\xi}\,\tilde{\phi}_\nu\,\partial_\lambda C_\xi - \left(\partial^\mu C^\nu - \partial^\nu C^\mu\right)B_\nu \mp \tfrac{1}{2}\,\lambda\,B^\mu \\
& \pm B\left(\partial^\mu C - m\,C^\mu\right) \pm \tfrac{1}{2}\,\rho\left(\partial^\mu\beta\right)\Big].
\end{aligned}
\tag{A18}
$$

Here, the superscript $(g)$ on the Lagrangian density $[\mathcal{L}_{\mathcal{B}}^{(g)}]$ denotes the generalized form of the Lagrangian density (49), where ($\pm$) signs are present at appropriate places.

However, we shall see that it is the *latter* Lagrangian density that satisfies all the essential features.

As a consequence of the above observation in (A18), it is clear that the action integral $S = \int d^4 x \, \mathcal{L}_{\mathcal{B}}^{(g)}$ remains invariant ($s_b \, S = 0$) under the infinitesimal, continuous, and off-shell nilpotent ($s_b^2 = 0$) BRST transformations (A16). A noteworthy point, at this juncture, is the observation that ($\pm$) signs, associated with ($\pm m \, \tilde{\phi}_\mu$, $\pm m \, \phi_\mu$) in the kinetic term and gauge-fixing term, respectively, have been changed to ($+m \, \tilde{\phi}_\mu$, $+m \, \phi_\mu$) because *only* this choice of sign is allowed by the (co-)BRST transformations (A19) (see below) and (A16), respectively. This generalized Lagrangian density ($\mathcal{L}_{\mathcal{B}}^{(g)}$) *also* respects a set of off-shell nilpotent ($s_d^2 = 0$) dual-BRST (i.e., co-BRST) symmetry transformations ($s_d$), as follows

$$
\begin{aligned}
s_d \, B_{\mu\nu} &= - \, \varepsilon_{\mu\nu\lambda\xi} \, \partial^\lambda \, \bar{C}^\xi, & s_d \, \bar{C}_\mu &= - \, \partial_\mu \, \bar{\beta}, & s_d \, C_\mu &= \mathcal{B}_\mu, \\
s_d \, \tilde{\phi} &= \mp \rho, & s_d \, \tilde{\phi}_\mu &= \pm \, (\partial_\mu \, \bar{C} - m \, \bar{C}_\mu), & & \\
s_d \, \beta &= \mp \lambda, & s_d \, C &= \mathcal{B}, & s_d \, \bar{C} &= - \, m \, \bar{\beta}, \\
s_d \, [(\partial^\nu \, B_{\nu\mu}), & \mathcal{B}_\mu, \, B_\mu, \, \mathcal{B}, \, \phi, \, \phi_\mu, \, \Phi_{\mu\nu}, \, \bar{\beta}, \, \lambda, \, \rho] &= 0,
\end{aligned}
\tag{A19}
$$

because $\mathcal{L}_{\mathcal{B}}^{(g)}$ transforms to a *total* spacetime derivative in 4D as follows:

$$
\begin{aligned}
s_d \, \mathcal{L}_{\mathcal{B}}^{(g)} = \partial_\mu \Big[ & \mp \, m \, \varepsilon^{\mu\nu\lambda\xi} \, \phi_\nu \, \partial_\lambda \, \bar{C}_\xi + (\partial^\mu \, \bar{C}^\nu - \partial^\nu \, \bar{C}^\mu) \, \mathcal{B}_\nu \mp \tfrac{1}{2} \, \rho \, \mathcal{B}^\mu \\
& \mp \mathcal{B} (\partial^\mu \, \bar{C} - m \, \bar{C}^\mu) \pm \tfrac{1}{2} \, \lambda \, (\partial^\mu \, \bar{\beta}) \Big].
\end{aligned}
\tag{A20}
$$

As a consequence, it is crystal clear that the infinitesimal, continuous and off-shell nilpotent ($s_d^2 = 0$) co-BRST transformations (A19) are the *symmetry* transformations for the action integral $S = \int d^4 x \, \mathcal{L}_{\mathcal{B}}^{(g)}$ due to the validity of the Gauss divergence theorem.

We comment on the fact that the modified SF (cf. Equation (9)) and Lagrangian density (cf. Equations (49) and (A17)) remain invariant under the discrete duality symmetry transformations (cf. Equations (48) and (54)) at the *quantum* level. Furthermore, these *latter* discrete symmetry transformations provide a connection between the BRST symmetry transformations (A16) and the co-BRST symmetry transformations (A19) in *exactly* the same manner as we have discussed such kind of relationship in the *simpler* case of Lagrangian density (49) in Section 6. To take a simple example, let us focus on $s_b \, \phi = \pm \, \lambda$. If we take the input $s_b \to s_d$ *and* the discrete symmetry transformations: $\phi \longrightarrow \pm i \, \tilde{\phi}$, $\lambda \longrightarrow \mp i \, \rho$ (cf. Equations (48) and (54)), we obtain $s_d \, \tilde{\phi} = \mp \rho$ from $s_b \, \phi = \pm \, \lambda$. This reciprocal relationship, it can be readily checked, is *also* true where we obtain $s_b \, \phi = \pm \, \lambda$ from $s_d \, \tilde{\phi} = \mp \rho$ if we take into account: $s_d \to s_b$ and the discrete symmetry transformations (48) and (54) *together*. In addition, we find that the algebraic relationship (cf. Equation (60)) is *also* satisfied by the (co-)BRST symmetry transformations (A19) and (A16), respectively. Corresponding to this observations, there is *also* an existence of the reciprocal relationship: $s_b = - \, [\pm * s_d *]$ where the symbols carry their standard meanings. Thus, we find that, as far as symmetry properties are concerned, we have obtained a generalized versions in (A16) and (A19), which are the symmetry transformations for the generalized version of the Lagrangian density $\mathcal{L}_{\mathcal{B}}^{(g)}$ [cf. Equation (A17)].

Despite the fact that we have the existence of (i) the generalized form of the Lagrangian density ($\mathcal{L}_{\mathcal{B}}^{(g)}$) and (ii) the generalized versions of the (co-)BRST symmetry transformations (A19) and (A16), respectively, we find that the equations of motion for the fermionic Lorentz vector (anti-)ghost fields do *not* obey the *normal* Klein–Gordon equations: $(\Box + m^2) \, \bar{C}_\mu = 0$ and $(\Box + m^2) \, C_\mu = 0$. However, it is interesting and gratifying to state that the Lagrangian density (49) respects (i) the (co-)BRST symmetry transformations (52) and (50), and (ii) the existence of the EL-EoMs: $(\Box + m^2) \, \bar{C}_\mu = 0$ and $(\Box + m^2) \, C_\mu = 0$ is also *true* provided we *also* use the EoMs w.r.t. the auxiliary fermionic fields $\lambda$ and $\rho$. Thus, the Lagrangian density (49) is *unique* in the sense that all its terms carry a definite sign which cannot be altered in any manner [10]. As a consequence, the form of the (co-)BRST symmetry transformations (52)

and (50) will also *not* change. It also respects the discrete duality symmetry transformations (48) *plus* (54). Similarly, one can have a Lagrangian density (cf. e.g., [10] for details) which respects the anti-BRST and anti-co-BRST symmetry transformations along with the discrete duality symmetry transformations (48) *plus* (54). Thus, we conclude that the Lagrangian density (49), which respects a set of *six* continuous and a couple of discrete duality symmetry transformations, is *unique*, and its symmetries and conserved charges provide the physical realizations of the de Rham cohomological operators of differential geometry at the *algebraic* level. Hence, as it turns out, the Stückelberg-modified *massive* 4D Abelian 2-form free theory becomes a tractable field-theoretic example for Hodge theory [10], which requires the incorporation of a set of *exotic* new fields with negative kinetic terms (but with a well-defined rest mass). The *latter* fields are found to be a set of possible candidates for the dark matter [19,20], and they *also* play crucial roles in explaining some of the theoretical issues connected with the cyclic, bouncing, and self-accelerated cosmological models of the Universe (see, e.g., [13–15]).

## Notes

[1]   We assume that the parity symmetry is respected in our discussion on the *massive* 4D Abelian 2-form theory (unlike the parity *violation* in the context of theoretical description of the weak interactions).

[2]   It can be seen that the equation of motion $\partial_\alpha \tilde{\Phi}^{\alpha\beta} + m^2 \tilde{\phi}^\beta = 0$ implies that $\partial \cdot \tilde{\phi} = 0$ for $m^2 \neq 0$ due to the antisymmetric ($\tilde{\Phi}^{\alpha\beta} = - \tilde{\Phi}^{\beta\alpha}$) property of the $\tilde{\Phi}^{\alpha\beta}$. As a consequence, we obtain $(\Box + m^2) \tilde{\phi}^\beta = 0$. We shall see that *this* EL-EoM emerges out from the properly gauge-fixed *final* Lagrangian density (29). It is interesting to point out the EL-EoMs: $(\Box + m^2) \tilde{\phi}_\mu = 0$ and $\partial^\nu \tilde{\Phi}_{\nu\mu} + m^2 \tilde{\phi}_\mu = 0$ are equivalent to each other as are the EL-EoMs for the antisymmetric tensor field $B_{\mu\nu}$: $(\Box + m^2) B_{\mu\nu} = 0$ and $\partial^\lambda H_{\lambda\mu\nu} + m^2 B_{\mu\nu} = 0$.

[3]   It is worthwhile to mention here that the kinetic terms for the $\phi_\mu$ and $\tilde{\phi}_\mu$ fields have a relative *sign* difference. In other words, one of the above fields has a *negative* kinetic ter,m which is *interesting*.

[4]   The gauge-fixing term of our Equation (30), with a pure scalar field $\phi$, can be found in the book by: M. Henneaux, C. Teitelboim, *Quantization of Gauge Systems*, Princeton University Press, Princeton, 1992. However, the nilpotent (anti-)BRST symmetries, discussed in this book, are *not* absolutely anticommuting in nature (see, e.g., [18] for details). The anticommuting property (i.e., one of the crucial requirements of the BRST formalism) is satisfied *only* up to a $U(1)$ gauge symmetry transformation.

[5]   We lay emphases on the fact that the axial-vector field ($\tilde{\phi}_\mu$) and pseudo-scalar field ($\tilde{\phi}$) possess *negative* kinetic terms. However, they satisfy the proper Klein–Gordon equation. Hence, these fields correspond to the *exotic* relativistic particles with well-defined rest *mass*. As a consequence, they are the possible candidates of dark matter [19,20] and they are also similar to the "phantom" and/or "ghost" fields in the realm of cosmology [13–15].

[6]   We stress the fact that the discrete *duality* symmetry transformations (33) and the Euler–Lagrange equations of motion (34) are *true* for *both* the Lagrangian densities $\mathcal{L}_{(1)}$ and $\mathcal{L}_{(2)}$ (cf. Equations (31) and (32)).

[7]   For the sake of brevity, we have taken *only* one specific sign in the kinetic energy and gauge-fixing terms for the $B_{\mu\nu}$ and associated fields. This is *true* for the (anti-)ghost fields, as well. In our Appendix B, we take the *most* general form of the (co-)BRST invariant Lagrangian density, which respects the generalized forms of (co-)BRST symmetry transformations corresponding to *classical* transformations (37) and (35).

[8]   It can be readily checked that the Faddeev–Popov ghost part of the Lagrangian density $\mathcal{L}_B$ remains invariant under a *couple* of discrete symmetry transformations hidden in Equation (54).

[9]   The ($\pm$) signs on the r.h.s. of (60) are dictated by the *successive* operations of the discrete duality symmetry transformations (48) and (54) on the generic field: $\Phi = B_{\mu\nu}, \phi_\mu, \tilde{\phi}_\mu, C_\mu, \bar{C}_\mu, \phi, \tilde{\phi},$, etc. In other words, the signs on the r.h.s. of $* (* \Phi) = \pm \Phi$ dictate the signs of Equation (60) (see, e.g., [5,10]).

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
