# Peer review of "Modified Stückelberg Formalism: Free Massive Abelian 2-Form Theory in 4D"

_universe, doi:10.3390/universe9040191_

Round 1
Reviewer 1 Report
The paper is very verbose -- the results do not seem to be incorrect, but they are buried under a lot of words and a lot of equations, most of which seem to be common to earlier works, maybe with minor modifications. Also, there are no observable effects, and the theory is not very complicated, as all the fields are boson fields and the continuous symmetry is Abelian.
Given this, I do not feel the paper justifies its length. It may be published, but only if the length is significantly reduced. In particular, there is no need to describe and explain every step in words, or to use so many adjectives.
Author Response
We have attached the appropriate PDF file.

Reviewer 2 Report
Referee report is attached.

Author Response

(The authors gave the same response as above.)

Reviewer 3 Report
The manuscript Modified Stückelberg Formalism: Free MassiveAbelian 2-Form Theory in 4D discusses a generalization of the old usual Stückelberg trick to obtain gauge invariance (introducing auxiliary fields) from a corresponding second class system in the realm of abelian p-form theories and proceeds with its (anti-)(co-)BRST quantization. As its title states, although the authors conjecture their results to be valid for free massive abelian p-form theories in 4p dimensions, in this specific manuscript they work out explicitly the case p=2, D=4, (for p=1, D=2 they have already demonstrated its validity in a previous article and for p=3, D=6 they mention a work in progress in the conclusion to appear in the future). The text is well written, interesting to read and the proposed generalization includes important new symmetries, techniques and results. However, there are a few not so clear points in the text, namely: 1) The authors suggest that the pseudo-scalar fields might be a candidate for dark matter/dark energy. It seems this argument is valid only for the precise dimension 4p, that is, D=2 for 1-form fields and D=4 for 2-form fields. Since we have different proposals for the number of space dimension coming from string theories other alternative models, the authors could include a short comment on that. 2) 191-193 -> I guess when the authors state that "the Stuckelberg modified Lagrangian density (LS) does not transform to a total spacetime derivative" they want to say that the Lagrangian density is invariant itself (before the integration in the action). Maybe the sentence could be clarified a bit more. 3) 307-309 -> It is interesting to see (as in eq (34) for instance) that all fields satisfy the Klein-Gordon equation with the same mass m. What is the physical interpretation of that? Does it mean all of them have the same mass? 4) 319-320 -> Eq. (30) does not make sense as it is. Maybe the authors want to say that somehow it is equivalent to introduce the Nakanish-Lautrup fields and obtain the same dynamics, or some functional integration within a generating functional is understood or similar. I understand that the concept/idea behind it maybe correct, but a mathematical statement such as eq (30) should be fully clarified. 5) It is stated on line 656 that eq (A.15) respects the field's dual discrete symmetry. That does not seem to be the case due to the gauge-fixing term which sends \phi to -+i\tilde{\phi}. 6) There are some recent relevant publications from other different authors on variations of (anti-)(co-)BRST symmetries which were not mentioned. For instance the work B. P. Mandal, S. K. Rai and R. Thibes, New Forms of BRST Symmetry on a Prototypical First-Class System, [arXiv:2210.15583 [hep-th]] containing novel BRST symmetries and a nice account on recent works on the field should be cited. The authors should carefully address the six points above mentioned with corresponding modifications on the text.
Author Response

(The authors gave the same response as above.)
